# The Performance Of The Unadjusted Langevin Algorithm Without Smoothness Assumptions

**Tim Johnston**                                      *timothy.johnston@dauphine.psl.eu*
*Ceremade*
*Université Paris Dauphine-PSL, France*

**Iosif Lytras**                                            *i.lytras@athenarc.gr*
*Archimedes*
*Athena Research Center, Greece*

**Nikolaos Makras**                                    *N.Makras@sms.ed.ac.uk*
*School of Mathematics*
*University of Edinburgh, United Kingdom*

**Sotirios Sabanis**                                      *S.Sabanis@ed.ac.uk*
*School of Mathematics*
*University of Edinburgh, United Kingdom*
*School of Applied Mathematical and Physical Sciences*
*National Technical University of Athens, Greece*
*Archimedes*
*Athena Research Center, Greece*

**Reviewed on OpenReview:** *https://openreview.net/forum?id=TTNeuyYdhg*

## Abstract

In this article, we study the problem of sampling from distributions whose densities are not necessarily smooth nor logconcave. We propose a simple Langevin-based algorithm that does not rely on popular but computationally challenging techniques, such as the Moreau-Yosida envelope or Gaussian smoothing, and show consequently that the performance of samplers like ULA does not necessarily degenerate arbitrarily with low regularity. In particular, we show that the Lipschitz or Hölder continuity assumption can be replaced by a geometric one-sided Lipschitz condition that allows even for discontinuous log-gradients. We derive non-asymptotic guarantees for the convergence of the algorithm to the target distribution in Wasserstein distances. Non-asymptotic bounds are also provided for the performance of the algorithm as an optimizer, specifically for the solution of associated excess risk optimization problems.

## 1 Introduction

Sampling from non-smooth potentials arises in various fields, including Bayesian inference with sparsity-promoting priors, non-smooth optimization problems, and constrained sampling in physics and computational statistics. Traditional Markov Chain Monte Carlo (MCMC) methods, such as the Metropolis-Hastings algorithm, often encounter difficulties in exploring distributions defined by non-differentiable energy functions due to their reliance on local gradient information for efficient proposal mechanisms. Langevin dynamics provides a natural framework for sampling from a target distribution $\pi_\beta(x) \propto e^{-\beta u(x)}$, where $u(x)$ is a potential function. Let $\theta_0$ be an $\mathbb{R}^d$-valued random variable, $\beta > 0$ the inverse temperature parameter, and $(B_t)_{t \geq 0}$ a $d$-dimensional Brownian motion. Given $Z_0 = \theta_0$, the overdamped Langevin equation

$$dZ_t = -\nabla u(Z_t)\,dt + \sqrt{2\beta^{-1}}dB_t, \ t \in [0, \infty) \tag{1}$$

drives a diffusion process whose stationary distribution matches $\pi_\beta(x)$. However, in non-smooth settings where $u(x)$ lacks differentiability, the gradient $\nabla u(x)$ may not be well-defined, leading to difficulties in simulating Langevin dynamics directly. Such challenges arise in problems involving $\ell_1$ regularization (as in LASSO), total variation priors, and energy-based models with discontinuous potentials. There has been a vast literature in sampling from non-smooth potentials through Langevin dynamics where people either use smoothing techniques such as the Moreau-Yosida envelope (Pereyra, 2016; Brosse et al., 2017; Durmus et al., 2022) or Gaussian smoothing (Chatterji et al., 2020; Laumont et al., 2022; Nguyen et al., 2023), or other more direct and computationally efficient methods such as (Lehec, 2023; Johnston & Sabanis, 2023; Habring et al., 2024; Fruehwirth & Habring, 2024). This topic is relevant for practitioners since it is known that loss landscapes in application are not necessarily smooth, see (Wang et al., 2023).

Despite extensive efforts in the field, our understanding of the literature remains primarily focused on the logconcave case, which leads to the following question that this work seeks to address rigorously:

*Can we design a simple, computationally efficient and explicit algorithm to sample from non-smooth non-logconcave distributions?*

This article advances the current state of the art in Langevin-based sampling from non-smooth potentials, extending the focus beyond logconcavity to encompass semi-logconcavity, by providing a simple, computationally efficient algorithm for which non-asymptotic convergence guarantees are obtained in Wasserstein distances.

As we gradually move towards potentials that are non-logconcave, a second challenge of this work is to establish connections with non-convex optimization in directions that are important for computational statistics, inverse problems, and machine learning. Intuitively, by the known fact that $\pi_\beta$ concentrates around the (global) minimizers of $u$ for large values of $\beta$, see (Hwang, 1980; Raginsky et al., 2017), it seems natural that our algorithm is well placed to solve (expected) excess risk optimization problems of the form $u(\hat\theta) - \inf_{\theta \in \mathbb{R}^d}(u(\theta))$, where $\hat\theta$ is an estimator of a global minimizer $\theta^*$. This leads us to a second challenge:

*Can this sampling algorithm perform as an optimizer in the associated expected excess risk optimization problem?*

To answer this question we produce a result of the form

$$\mathbb{E}[u(\theta_n^\lambda)] - u(\theta^*) \leq C \left( W_2(\mathcal{L}(\theta_n^\lambda), \pi_\beta) + \beta^{-1} \log(\beta) \right),$$

where $C$ is independent of the variables under discussion and $(\theta_n^\lambda)_{n \geq 0}$ denotes the iterates of our proposed algorithm. Moreover, the first term is controlled by the sampling guarantees of our algorithm, while the second term decays for large $\beta$. Our approach combines new findings in non-smooth, non-logconcave sampling with expected excess risk estimates, thereby presenting the first such contribution in the Langevin-based sampling literature for non-smooth potentials.

## 1.1 Related Literature

Throughout the last decade there has been a remarkable progress in the field of sampling with Langevin-based algorithms. The vast majority of the literature deals with potentials that are differentiable and can be categorized with respect to gradient smoothness.

### 1.1.1 Results for potentials with Lipschitz-smooth gradients

This assumption is ever present in the literature in a great deal of works. Under the assumption of convexity and gradient Lipschitz continuity important results are obtained in (Dalalyan, 2017; Durmus & Moulines, 2017; 2019; Sabanis & Zhang, 2019; Barkhagen et al., 2021), while in the non-convex case, under convexity

at infinity or dissipativity assumptions, one may consult Cheng et al. (2018); Majka et al. (2020); Erdogdu et al. (2022) for ULA while for the Stochastic Gradient variant (SGLD) important works are (Raginsky et al., 2017; Chau et al., 2021; Zhang et al., 2023b). More recently, starting with the work of Vempala & Wibisono (2019) important estimates have been obtained under the assumption that the target measure $\pi_\beta$ satisfies an isoperimetric inequality see, (Mou et al., 2022; Erdogdu et al., 2022; Chewi et al., 2024).

### 1.1.2   Results for non-Lipschitz smooth gradients

Recently there has been a lot of effort in exploring settings beyond Lipschitz gradient continuity.

**(A) Locally Lipschitz gradients**
In the case of locally Lipschitz gradients (where the gradient is allowed to grow superlinearly) there has been important work using Langevin algorithms based on the taming technique starting with (Brosse et al., 2019; Johnston et al., 2024), for strongly convex potentials, while in the non-convex case key references are (Neufeld et al., 2025) using a convexity at infinity assumption, (Lytras & Sabanis, 2025; Lytras & Mertikopoulos, 2025) for results under the assumption of a functional inequality and (Lovas et al., 2023; Lim & Sabanis, 2024) for results involving stochastic gradients.

**(B) Hölder continuous gradients**
In order to deal with potentials with thin tails, recently, there has been a lot of effort to relax the gradient Lipschitz continuity assumption with a Hölder one. The first results were obtained under a dissipativity assumption in (Erdogdu & Hosseinzadeh, 2021; Nguyen et al., 2023) which was later dropped to provide results in Rényi divergence under relaxed conditions in (Chewi et al., 2024; Mousavi-Hosseini et al., 2023) under a Poincaré and weak Poincaré inequality and for the underdamped Langevin algorithm in (Zhang et al., 2023a). However all these results degenerate with the Hölder regularity of the coefficients, that is the upper bound on the algorithm is arbitrary large in the low regularity case. We show using Assumption A4 that the Hölder regularity condition can be replaced by a geometric condition, and that one obtains explicit sampling bounds even in the case of discontinuous log-gradients in a non-convex setting.

### 1.1.3   Results for non-smooth potentials

Sampling from densities where the potential is not differentiable is a very prominent problem with both theoretical and practical interest for fields like inverse problems and Bayesian inference. Classical example in statistics is regression with Lasso priors or $L_1$-loss and non-smooth regularization functionals in Bayesian imaging. Consequently, since vanilla ULA relies on access to the gradient of the potential, which does not exist in the non-smooth setting, there is a significant gap in the current theory that remains to be addressed. To tackle this problem, two main approaches have been used so far: subgradient algorithms and smoothing techniques.

**(A) Smoothing techniques**
Smoothing techniques have been the go-to methodology for the majority of works. The earliest contributions in this direction applied the Moreau-Yosida ULA (MYULA) framework, as reported in (Pereyra, 2016; Brosse et al., 2017; Durmus et al., 2022). The algorithm is based on the use of the Moreau-Yosida envelope. Essentially, one first samples from an approximating measure that has a Lipschitz-smooth log-gradient and then connects it with the original target measure. Important results have been obtained in total variation. Extensions of these works have been incorporating Metropolis steps resulting in the Proximal Metropolis Langevin Algorithm, see (Cai et al., 2022; Pereyra, 2016). Although these works have achieved rigorous results, their main drawback is the added computational burden at each iteration due to the computation of the MY envelope. Efforts to reduce this computational cost have been made through inexact proximal mapping in (Ehrhardt et al., 2024), where the results are limited to the class of logconcave distributions. Another popular smoothing technique involves smoothing the density by applying the Gaussian kernel to the subgradients and sampling from the smoothed potential, which approximates the original target, see (Chatterji et al., 2020; Laumont et al., 2022; Nguyen et al., 2023). A drawback of these interesting results

**Table 1:** Comparison of algorithmic complexity across existing literature.
Abbreviations: C – convex, SC – strongly convex, WC – weakly convex, LG – linear growth, B – bounded.

| | $W_1$ | $W_2$ | $KL$ | $TV$ | CONVEXITY | SUBGRADIENT |
|---|---|---|---|---|---|---|
| Lehec (2023) | - | $\Theta\left(\epsilon^{-2}\right)$ | - | - | C | LG |
| Johnston & Sabanis (2023) | - | $\Theta\left(\epsilon^{-4}\right)$ | - | - | SC | LG |
| Habring et al. (2024) | - | - | $\Theta\left(\epsilon^{-3}\right)$ | $\Theta\left(\epsilon^{-6}\right)$ | C | B |
| Fruehwirth & Habring (2024) | - | $\Theta\left(\epsilon^{-2}\right)$ | - | - | SC | LG |
| Present work (under A2) | $\Theta\left(\epsilon^{-4}\right)$ | $\Theta\left(\epsilon^{-8}\right)$ | - | - | WC | LG |
| Present work (under A5) | $\Theta\left(\epsilon^{-2}\right)$ | $\Theta\left(\epsilon^{-4}\right)$ | - | - | WC | LG |

is that they are obtained under additional smoothness assumptions for the gradients and also increase the computational burden at each iterate.

**(B) Subgradient algorithms**
Another important class of algorithms involves the use of subgradients. Initial progress was made by (Durmus & Moulines, 2019) which was subsequently adapted to achieve improved convergence results in (Habring et al., 2024). The work in (Fruehwirth & Habring, 2024) weaken the required assumptions by permitting linear growth of the subgradient, extending the previous framework where the subgradient was assumed to be the sum of a Lipschitz function and a globally bounded coefficient. Under similar assumptions in the logconcave case, the work in (Lehec, 2023) serves as another key reference, where the authors first derive results for constrained sampling, yielding results for logconcave measures with full support. In parallel to these developments, (Johnston & Sabanis, 2023) obtained related results, establishing Wasserstein-type bounds under either piecewise Lipschitz continuity or linear growth. Their analysis, however, requires strongly convex potentials. Although substantial progress has been made on sampling from non-smooth potentials, the non-convex setting remains comparatively less explored.

### 1.1.4 Euler Scheme Approximations

Recently, significant progress has been made in the numerical analysis literature on the subject of SDEs with discontinuous drift coefficient, see (Müller-Gronbach & Yaroslavtseva, 2024) for a survey. In particular, the performance of the Euler scheme with discontinuous coefficients was investigated in (Müller-Gronbach & Yaroslavtseva, 2020; Dareiotis et al., 2023) and many others, and lower bounds established in (Hefter et al., 2019; Ellinger, 2024).

### 1.2 Summary of contributions and comparison with literature

This article aims to expand the state of the art in Langevin-based sampling from non-smooth potentials beyond logconcavity, specifically to semi-logconcavity (for the rigorous definition see (Cattiaux & Guillin, 2014)), and to establish connections with non-convex optimization. The contributions of our work can be summarized as follows:

- We provide rigorous results for the treatment of SDEs with discontinuous drifts beyond logconcavity.

- For stepsize $\lambda$, we achieve $\lambda^{1/4}$ rates in $W_1$ distance and $\lambda^{1/8}$ in $W_2$ distance for our algorithm to the target measure. To the best of our knowledge, these are the first results under such weak assumptions.

- We utilize these findings to derive explicit bounds for the associated (expected) excess risk optimization problem, thereby presenting the first such contribution in the Langevin-based sampling literature for non-smooth potentials.

The following table compares the performance of our algorithm with methods that do not rely on smoothing techniques, thereby avoiding the additional complexity such techniques introduce. Our results compare favorably with the state of the art, although the rate of convergence is influenced by the fact that the analysis is carried out in a non-convex setting. The presence of non-convexity prevents the use of certain tools, such as the $W_1$-$TV$ relations developed in (Lehec, 2023) or the $W_2$-$KL$ connections used in the convergence proof in (Fruehwirth & Habring, 2024). Another important element when comparing with (Fruehwirth & Habring, 2024) is the fact that our constants are explicit.

## 1.3 Notation

We introduce some basic notation. For $x, y \in \mathbb{R}^d$, define the scalar product $\langle x, y \rangle = \sum_{i=1}^{d} x_i y_i$ and the Euclidian norm $|x| = \sqrt{\langle x, x \rangle}$. For all continuously differentiable functions $f : \mathbb{R}^d \to \mathbb{R}$, $\nabla f$ denotes the gradient. The integer part of a real number $x$ is denoted by $\lfloor x \rfloor$. For an $\mathbb{R}^d$-valued random variable $Z$, its law on $\mathcal{B}(\mathbb{R}^d)$, i.e. the Borel sigma-algebra of $\mathbb{R}^d$, is denoted by $\mathcal{L}(Z)$. We denote by $\mathcal{P}(\mathbb{R}^d)$ the set of all probability measures on $\mathcal{B}(\mathbb{R}^d)$ and for any $p \in \mathbb{N}$, $\mathcal{P}_p(\mathbb{R}^d) = \{\pi \in \mathcal{P}(\mathbb{R}^d) : \int_{\mathbb{R}^d} |x|^p d\pi(x) < \infty\}$ denotes the set of all probability measures over $\mathcal{B}(\mathbb{R}^d)$ with finite $p$-th moment. For any two probability measures $\mu$ and $\nu$, we define the Wasserstein distance of order $p \geq 1$ as

$$W_p(\mu, \nu) = \left( \inf_{\zeta \in \prod(\mu, \nu)} \int_{\mathbb{R}^d \times \mathbb{R}^d} |x - y|^p d\zeta(x, y) \right)^{1/p},$$

where $\prod(\mu, \nu)$ is the set of all transference plans of $\mu$ and $\nu$. Moreover, for any $\mu$, $\nu \in \mathcal{P}_p(\mathbb{R}^d)$, there exists a transference plan $\zeta^* \in \prod(\mu, \nu)$ such that for any coupling $(X, Y)$ distributed according to $\zeta^*$, $W_p(\mu, \nu) = \mathbb{E}^{1/p}[|X - Y|^p]$.

## 2 The Non-Convex Setting

### 2.1 Subdifferentiability for non-smooth functions - subgradients

Given that the potentials discussed in this article are non-smooth, it is natural to describe them using the concept of subdifferentials. For any $x \in \mathbb{R}^d$ and any $u : \mathbb{R}^d \to \mathbb{R}$, the subdifferential $\partial u(x)$ of $u$ at $x$ is defined by

$$\partial u(x) := \left\{ p \in \mathbb{R}^d : \liminf_{y \to x} \frac{u(y) - u(x) - \langle p, y - x \rangle}{|y - x|} \geq 0 \right\}.$$

The subdifferential is a closed convex set, possibly empty. If $u$ is a convex function, the above set coincides with the well-known subdifferential of convex analysis, which captures all relevant differential properties of convex functions. Similar nice properties exist in the case of a larger class of functions, namely the class of semi-convex functions.

**Definition 1.** Let $u : \mathbb{R}^d \to \mathbb{R}$, we say that $u$ is $K$-semi-convex if and only if there exists $K \geq 0$ such that the function $x \to u(x) + \frac{K}{2}|x|^2$ is convex.

**Lemma 1** ((Alberti et al., 1992), Proposition 2.1, adapted)**.** *Let $u$ be a semi-convex function. Then, $u$ is locally Lipschitz continuous, the sets $\partial u(x)$ are non-empty, compact, and $p \in \partial u(x)$, if and only if*

$$u(y) - u(x) - \langle p, y - x \rangle \geq -\frac{K}{2}|y - x|^2 \quad \forall x, y \in \mathbb{R}^d.$$

**Corollary 1.** *Let $x, y \in \mathbb{R}^d$, $p \in \partial u(x)$ and $q \in \partial u(y)$. Then,*

$$\langle p - q, x - y \rangle \geq -K|x - y|^2.$$

At the points where $u$ is differentiable it holds that, $\partial u(x) = \{\nabla u(x)\}$. From these results, one can see that the class of semi-convex functions is an ideal starting point to proceed from convexity to non-convexity, as all the elements of the subdifferential set satisfy an one-sided Lipschitz continuity property.

## 2.2 Assumptions

For clarity and brevity reasons, it is assumed that, henceforth, $h(x)$ denotes an element of $\partial u(x)$, for any $x \in \mathbb{R}^d$. We proceed with our main assumptions.

**Assumption 1.** *The gradient of $u$ exists almost everywhere and each subgradient grows at most linearly. That is, there exist $L, m > 0$, such that for each subgradient $h \in \partial u$*

$$|h(x)| \leq m + L|x|, \ \forall x \in \mathbb{R}^d. \tag{2}$$

This assumption allows the use of explicit numerical algorithms based on popular discretization schemes such as Euler-Maruyama, and is on par with the weakest assumptions (in the presence of discontinuous) in the related literature.

**Assumption 2.** *The potential is strongly convex at infinity (outside a compact set). That is, there exist $\mu > 0$ and $R > 0$, such that, for $x, y \in \mathbb{R}^d$,*

$$\langle h(x) - h(y), x - y \rangle \geq \mu|x - y|^2, \quad if \ |x - y| \geq R. \tag{3}$$

Assumption A2 is essentially a geometric condition that is crucial for obtaining contraction results in Wasserstein distances. Furthermore, combined with Assumption A1, they yield the following dissipativity property:

**Lemma 2.** *Let Assumptions A1 and A2 hold, then $h$ is dissipative. That is, there exists $b > 0$, such that*

$$\langle x, h(x) \rangle \geq \frac{\mu}{2}|x|^2 - b, \ \forall x \in \mathbb{R}^d. \tag{4}$$

*Proof.* The proof is postponed to Appendix G. $\qquad\square$

This is a growth condition that guarantees uniform control of (polynomial) moments for both the proposed (explicit) algorithm and for the associated Langevin stochastic differential equation.

**Assumption 3.** *The initial condition of the algorithm is an $\mathbb{R}^d$-valued random variable with finite 2nd moment, i.e.*

$$\mathbb{E}|\theta_0|^2 < \infty. \tag{5}$$

**Assumption 4.** *The potential $u$ is $K$- semi-convex. That is, there exists $K \geq 0$, such that $u + \frac{K}{2}|\cdot|^2$ is convex. Due to Corollary 2.1, the following equivalent property for the subgradient holds*

$$\langle h(x) - h(y), x - y \rangle \geq -K|x - y|^2, \ \forall x, y \in \mathbb{R}^d. \tag{6}$$

The last of our main assumptions, Assumption A4, characterizes the lack of smoothness for the subgradients in our article. Assumption A4 is geometric in nature, and is often referred to as a 'one-sided Lipschitz assumption'. This suggests that algorithm performance is not hindered by regularity in the way often suggested in the literature, and that 'bad' regularity in 'some' directions does not necessarily hinder performance arbitrarily (see Section 1.1.2 for more details). It is key to our approach in proving contraction estimates necessary for solving associated sampling and (possibly non-convex) optimization problems. In essence, A2 ensures that any two sufficiently separated trajectories are, on average, driven together, whereas A4, which provides a lower bound on the local negative curvature, guarantees that once they are close they cannot separate too aggressively.

## 3 Main Results

The Subgradient Unadjusted Langevin Algorithm $(\theta_n^\lambda)_{n\geq 0}$, is given by the Euler-Maruyama discretisation scheme of (1), in particular

$$(\textbf{SG-ULA}): \qquad \theta_{n+1}^\lambda = \theta_n^\lambda - \lambda h(\theta_n^\lambda) + \sqrt{2\lambda\beta^{-1}}\xi_{n+1}, \ \theta_0^\lambda = \theta_0, \ n \in \mathbb{N}, \qquad (7)$$

where $\lambda > 0$ is the stepsize of the algorithm, $\beta > 0$ is the inverse temperature parameter, $(\xi_n)_{n\geq 1}$ is a sequence of i.i.d. standard Gaussians on $\mathbb{R}^d$ and $h(x) \in \partial u(x)$, for all $x \in \mathbb{R}^d$. One can easily observe that the algorithm is an Euler-Maruyama discretization of a Langevin SDE with drift coefficient an element of the subdifferentials. One can further understand that the newly proposed algorithm is significantly easier to implement than popular algorithms which rely on smoothing techniques, such as MYULA, since these smoothing procedures increase the computational cost per iteration. Furthermore, our algorithm offers a generalization of ULA (since $u$ is assumed to be differentiable almost everywhere) and coincides with ULA when $u$ is continuously differentiable. To address the issue of having differentiability almost everywhere (and not for every $x \in \mathbb{R}^d$), we choose a subgradient for every $x \in \mathbb{R}^d$ and thus define $h$ for all $x \in \mathbb{R}^d$. Note again that at the points where $u$ is differentiable it holds that $\partial u(x) = \{\nabla u(x)\}$.

### 3.1 Theoretical Guarantees

**Theorem 1.** *Let Assumptions A1-A4 hold and set $\lambda_0 = \min\{\mu/(2L^2), 1\}$. For any $\lambda \in (0, \lambda_0)$ and $N \in \mathbb{N}$, the subgradient unadjusted Langevin algorithm (SG-ULA) given in (7) satisfies, for each $p \in \{1, 2\}$,*

$$W_p\left(\mathcal{L}(\theta_N^\lambda), \pi_\beta\right) \leq C_{W_p} e^{-C_{r_p}\lambda N}\Delta_0^{(p)} + C_{T_p}\lambda^{1/(4p)}. \qquad (8)$$

*The initialization terms are $\Delta_0^{(1)} = W_1(\mathcal{L}(\theta_0), \pi_\beta)$, $\Delta_0^{(2)} = \max\{W_2(\pi_\beta, \mathcal{L}(\theta_0)), W_1(\pi_\beta, \mathcal{L}(\theta_0))^{1/2}\}$. The constants $C_{W_p} = \mathcal{O}(\beta^{1-p}e^{R^2\beta^{(p+1)/2}})$ and $C_{r_p} = \mathcal{O}(\beta^{1-p/2}e^{-R^2\beta^{(p+1)/2}})$ do not depend on the dimension $d$, and $C_{T_p} = \mathcal{O}(d)$.*

*Proof.* The proof is postponed to Appendix E. $\qquad\square$

As discussed in Remark 3, the dependence on the stepsize can be improved to $\lambda^{1/(2p+2p\epsilon)}$ in Theorem 1, for any $\epsilon > 0$, at the expense of a stronger dependence of the constant on the dimension.

**Corollary 2.** *Let $\epsilon > 0$. Then, for $\lambda < \min\{\lambda_0, \frac{\epsilon^4}{16C_{T_1}^4}\}$, one needs $N \geq \mathcal{O}(\epsilon^{-4}16C_{T_1}^4 C_{r_1}^{-1}\log(2C_{W_1}\Delta_0^{(1)}/\epsilon))$ iterations to achieve*

$$W_1(\mathcal{L}(\theta_N^\lambda), \pi_\beta) \leq \epsilon.$$

**Corollary 3.** *Let $\epsilon > 0$. Then, for $\lambda < \min\{\lambda_0, \frac{\epsilon^8}{16^2C_{T_2}^8}\}$, one needs $N \geq \mathcal{O}(\epsilon^{-8}16^2C_{T_2}^8 C_{r_2}^{-1}\log(2C_{W_2}\Delta_0^{(2)}/\epsilon))$ iterations to achieve*

$$W_2(\mathcal{L}(\theta_N^\lambda), \pi_\beta) \leq \epsilon.$$

The above results exhibit a mild dependence on the dimension. This arises because (3) and (6) yield dimension free contraction estimates, while the remaining dimensional dependence enters through the moment bounds obtained via dissipativity (Lemma2) and linear growth (A1). The bounds depend on the constants $\beta$, $\mu$, $K$ and $R$, this reflects an inherent limitation of analyses in non-convex settings, where such constants cannot, in general, be avoided. Essentially the magnitude of $R$ and $K$ quantify the size of the region, where the potential $u$ exhibits non-convex behavior. All the constants appearing in Theorem 1 are given explicitly in Proposition 4 and 5, a summary can be found in Table 2.

By enforcing slightly stronger assumptions, the convergence rate in $W_2$ with respect to the stepsize can be improved.

**Assumption 5.** *There exist $R > 0$ and $\mu > 0$, such that for any $x \in \mathbb{R}^d$ with $|x| \geq R$,*

$$\langle h(x) - h(y), x - y\rangle \geq \mu|x - y|^2, \quad \forall y \neq x.$$

*In addition, following Monmarché (2023), we pose the following restriction on $\beta$:*

$$\beta \leq \frac{\mu d}{2K + \mu} \frac{1}{(K + \mu/4)R_*^2 + 2 \sup\{-\langle x, h(x)\rangle, |x| \leqslant R_*\}}, \tag{9}$$

*where $R_* = R(2 + 2K/\mu)^{1/d}$.*

**Remark 1.** *A simple example of a function in this regime is $u(x) = |x|^2 + f + g$ where $f$ is compactly supported in $\bar{B}(0,1)$ with a gradient that is $\frac{1}{2}$-Lipschitz, and $g$ is convex (possibly non-differentiable). The convexity at infinity condition is satisfied with $\mu = \frac{1}{2}$. For $d$ big enough, (9) is satisfied for many choices of $\beta$ and especially for $\beta = 1$, making the example relevant for sampling.*

**Theorem 2.** *Let Assumptions A1, A3, A4 and A5 hold. Let $N \in \mathbb{N}$. Then, for every $\lambda \in (0, \lambda_0)$, the subgradient unadjusted Langevin algorithm (SG-ULA) given in (7) satisfies*

$$W_2(\mathcal{L}(\theta_N^\lambda), \pi_\beta) \leq C_{W_2}^* e^{-C_{r_3}\lambda N} W_2(\pi_\beta, \mathcal{L}(\theta_0)) + C_{T_3}\lambda^{1/4},$$

*where $C_{r_3}$ is independent of the dimension, $C_{T_3} = \mathcal{O}(d)$ and $C_{W_2}^* = 1 + \mathcal{O}(d^{-1})$.*

*Proof.* The proof is postponed to Appendix E. $\qquad\square$

**Corollary 4.** *Let $\epsilon > 0$. Then, for $\lambda < \min\{\lambda_0, \frac{\epsilon^4}{16 C_{T_3}^4}\}$, one needs $N \geq \mathcal{O}\left(\epsilon^{-4} 4 C_{T_3}^4 C_{r_3}^{-1} \log(2C_{W_2}^* \Delta_0^{(3)}/\epsilon)\right)$ iterations to achieve*

$$W_2\left(\mathcal{L}(\theta_N^\lambda), \pi_\beta\right) \leq \epsilon.$$

One further notes that the proofs of the contraction theorems employed, along with our proof roadmap demonstrating convergence to the algorithm, rely on Grönwall-type arguments, which leads to an exponential dependence on these parameters.

We also show that the algorithm can serve as an optimizer for associated excess-risk problems. The result is aligned with Raginsky et al. (2017), but the proof differs significantly because of the non-Lipschitz setting.

**Theorem 3.** *Let Assumptions A1-A4 hold hold and $\lambda_0 = \min\{\mu/(2L^2), 1\}$. Then, for any $\beta \geq \max\{4/\mu, M^{-1}\}$, $\lambda \in (0, \lambda_0)$ and $n \in \mathbb{N}$, the following bound holds*

$$\mathbb{E}[u(\theta_n^\lambda)] - u(\theta^*) \leq C_{\mathcal{T}_1} W_2\left(\mathcal{L}(\theta_n^\lambda), \pi_\beta\right) + \frac{d}{2\beta}\log\left(\frac{2e(b + d/\beta)\beta^2 M^2}{\mu d}\right) + \frac{d}{\beta}\log(\beta M) - \frac{1}{\beta}\log(S_d/d) + \frac{2}{\beta},$$

*where $C_{\mathcal{T}_1} = \mathcal{O}(d^{1/2})$, $S_d = 2\pi^{d/2}\Gamma^{-1}(d/2)$ and $M = m + 3L/2 + L\sqrt{b/(2\mu)}$.*

*Proof.* The proof is postponed to Appendix E. $\qquad\square$

Interpreting Theorem 3, one notes that for sufficiently large $\beta$ the last four terms in the bound become negligible. One then selects the stepsize and number of iterations, following Corollary 3 to control the remaining sampling term. In most applications it is advantageous to take $\beta$ as large as permitted, so that the $W_2$ error is effectively governed by the bound in Theorem 1 rather than Theorem 2.

## 3.2 Overview of proof techniques

One needs to first show existence and uniqueness of the solution to the SDEs (Proposition 1) and also establish that the invariant measure exists, is unique (Proposition 2) and corresponds to $\pi_\beta$ (Proposition 3). This is achieved by adapting standard Lyapunov arguments to show tightness of the measures while the uniqueness is established by the contraction results for $W_1$ and $W_2$ Wasserstein distances. These results are key elements of our work which enable us to show the convergence of our algorithm to the target measure. In a nutshell our proof roadmap can be summarized as follows:

- By making use of the fact of the convexity outside of a ball property (which yields dissipativity) and the subgradient linear growth property, we are able to provide uniform, in the number of iterations, moment bounds for the algorithm (which are independent of the step-size), Lemma 4.

- We introduce an auxiliary process (Definition 2) which is a Langevin SDE with initial condition a previous iteration of the algorithm for which we able to derive moment bounds, Lemma 6.

- We control both the $W_1$ and $W_2$ distance between the auxiliary process and the continuous time interpolation of the algorithm. To obtain this result, the one-sided Lipschitz property of the drift coefficient (which follows from the semi-convexity of the potential) is key to permit the application of Grönwall-type estimates, and along with the uniform control of the moments and the linear growth of the drift, enable us to obtain $\lambda^{1/4}$ rates for $W_1$ and $W_2$ distances (Lemma 7).

- The contraction theorems for $W_1$ and $W_2$ enable us to control the Wasserstein distance between the auxiliary process and its corresponding Langevin SDE, by starting from the same initial condition given by the interpolated scheme of the algorithm (Lemmata 8, 9, 10).

- The final bound is established by the convergence to the Langevin SDE to the invariant measure.

To obtain the result for the (expected) excess risk optimization problem, we split the difference in the following way

$$\mathbb{E}[u(\theta_n^\lambda)] - u(\theta^*) = \left(\mathbb{E}[u(\theta_n^\lambda)] - \mathbb{E}[u(\theta_\infty)]\right) + \left(\mathbb{E}[u(\theta_\infty)] - u(\theta^*)\right),$$

where $\mathcal{L}(\theta_\infty) = \pi_\beta$. For the first term, we use a fundamental theorem of calculus (which can be applied since $u$ is differentiable a.s) and we are able to derive a term that is proportional to the $W_2$ distance between the algorithm and the target measure (Lemma 11). For the second term, we make use of the fact that $\pi_\beta$ concentrates around the minimizers of $u$ for large $\beta$. More specifically, we simplify the difference to an integral of the exponential distribution and then use standard concentration inequalities to complete the proof (Lemma 12).

## 4 Examples and Numerical Experiments

### 4.1 Mixture of Gaussians with an $L^1$-Laplacian prior

Consider a target distribution given by a mixture of Gaussians (MoG) likelihood with $K \in \mathbb{N}$ components and an isotropic Laplace prior on the entire data vector $x$, i.e. a prior density $\propto \exp(-\alpha|x|_1)$ where $|x|_1 = \sum_{i=1}^d |x_i|$. The unnormalized density is

$$\pi(x) \propto \left(\sum_{j=1}^K w_j \frac{1}{(2\pi\sigma_j^2)^{d/2}} \exp\left(-\frac{|x-\mu_j|^2}{2\sigma_j^2}\right)\right) \exp(-\alpha|x|_1), \ x \in \mathbb{R}^d, \tag{10}$$

with $\sigma_j > 0$, $\mu_j \in \mathbb{R}^d$, and $w_j \in [0,1]$, for $j \in \{1,\ldots,K\}$, such that $\sum_{j=1}^K w_j = 1$. The gradient of the corresponding potential (negative log-density) can be written as

$$\nabla u(x) = \frac{\sum_{j=1}^K \frac{w_j(x-\mu_j)}{\sigma_j^2(2\pi\sigma_j^2)^{d/2}} \exp\left(-\frac{|x-\mu_j|^2}{2\sigma_j^2}\right)}{\pi(x)} + \alpha\frac{x}{|x|_1}. \tag{11}$$

This gradient exhibits at most linear growth, which arises from the linear factors inside the sum of the numerator in the first term. Additionally, it is non-smooth due to the non-differentiability of the Laplacian prior. The corresponding subgradient $\partial u(x)$ has linear growth while being semi-convex and strongly convex at infinity. In particular, the MoG term is semi-convex and strongly convex at infinity, and the inclusion of the convex $L^1$ prior preserves those properties, due to being convex. Gentiloni-Silveri & Ocello (2025) provide a rigorous verification of these properties within the context of mixture models (see their Assumption H1 and Appendix A.1), thereby confirming that the MoG with a Laplacian prior potential satisfies the assumptions

of our framework. This model has indeed been studied in prior works on Langevin based algorithms, for example, Lau et al. (2024) consider it a representative non-logconcave, non-smooth target (see Sections 2.1 and 6.3), using it to evaluate ULA-type samplers under minimal assumptions.

### 4.1.1 Sampling

We compare SGULA and MYULA on the task of sampling from a two dimensional mixture of Gaussians with a Laplacian prior, in order to assess their empirical behavior in a non-convex, non-smooth setting. For fixed scale parameter $a = 0.15$, two mixtures configurations are considered with different number of components. The first comprises three components ($K = 3$), with weights $w = \{0.3, 0.4, 0.3\}$, mean vectors $\mu = \{(-2.6, 2.8), (0, 0), (2.2, -2.2)\}$ and isotropic variances $\sigma^2 = \{0.60, 0.80, 0.70\}$. The second model contains five components ($K = 5$), with weights $w = \{0.18, 0.22, 0.20, 0.22, 0.18\}$, mean vectors $\mu = \{(-3.0, 2.8), (-1.2, 0.8), (0.8, -0.4), (2.2, -2.0), (3.2, 2.4)\}$ and isotropic variances $\sigma^2 = \{0.55, 0.65, 0.50, 0.70, 0.60\}$.

Both samplers were implemented with a fixed stepsize $\lambda = 10^{-3}$ and inverse temperature parameter $\beta = 1$, and MYULA employed the same value for its smoothing parameter ($\gamma = \lambda$). For each method, we initialized 12 parallel chains from a broad uniform distribution on

$$[\min_j \mu_j - 2 \max_j \sigma_j^2, \max_j \mu_j + 2 \max_j \sigma_j^2]^2,$$

thereby obtaining over dispersed initial states suitable for evaluating cross modal mixing. Each chain was run for $52 \times 10^3$ iterations, discarding the first $12 \times 10^3$ as burn-in and retaining the rest for the assessment. Figure 1 compares the empirical densities obtained from pooled samples across all chains with the analytical ground truth densities. Each empirical density is estimated using a Gaussian kernel with Silverman's bandwidth, providing a smooth visualization of the sampling behavior across modes.

Both SGULA and MYULA recover the main structural features of the target density. All modes are identified, and the overall allocation of probability mass across regions is consistent with the true density. SGULA produces contours that align more closely with the circular geometry of the Gaussian components, indicating reduced smoothing bias. MYULA, while accurately capturing the dominant regions, exhibits higher concentration around central modes, suggesting more limited mode exploration. Notably, these results demonstrate that SGULA remains effective even in this non-convex and non-smooth setting, highlighting its robustness when sampling from complex posteriors. Additional experiments illustrating the effect of stepsize and inverse temperature on SGULA's performance are provided in Appendix H.

## 4.2 One-dimensional example satisfying the assumptions

Let $u : \mathbb{R} \to \mathbb{R}$ be a continuous function such that

$$u(x) = u_1(x) + u_2(x) + u_3(x), \quad \forall x \in \mathbb{R},$$

where $u_1$ is a (continuous) strongly convex function (on $\mathbb{R}$) with $h_1 := \nabla u_1$, $u_2$ is a continuously differentiable function with a Lipschitz continuous derivative $h_2 := \nabla u_2$ and $u_3$ is a continuous function with a non-decreasing, discontinuous derivative $h_3 := \partial u_3$. Thus, $\forall x, y \in \mathbb{R}$,

$$\exists \, \mu_1 > 0 \text{ such that } \langle h_1(x) - h_1(y), x - y \rangle \geq \mu_1 |x - y|^2,$$
$$\exists \, K_2 > 0 \text{ such that } |h_2(x) - h_2(y)| \leq K_2 |x - y|,$$
$$\text{and } \langle h_3(x) - h_3(y), x - y \rangle \geq 0.$$

Note that in higher dimensions, the properties for $h_1$, $h_2$ and $h_3$ also yield the desired result provided that convexity at infinity is also achieved. Furthermore, one trivially concludes, $\forall x, y \in \mathbb{R}$

$$\langle h(x) - h(y), x - y \rangle \geq (\mu_1 - K_2) |x - y|^2 \geq -K_2 |x - y|^2.$$

For a concrete example, we may use, $\forall x \in \mathbb{R}$

$$u_1(x) = 2(x + 3)^2 - 1/2,$$
$$u_2(x) = -8x^2 \, \mathbb{I}_{\{0 < x < 2\}} - 8x - 32(x - 1) \, \mathbb{I}_{\{x \geq 2\}},$$
$$u_3(x) = 10(x - 1)^8 \, \mathbb{I}_{\{1 < x < 2\}} + x + 90(x - 17/9) \, \mathbb{I}_{\{x \geq 2\}}.$$

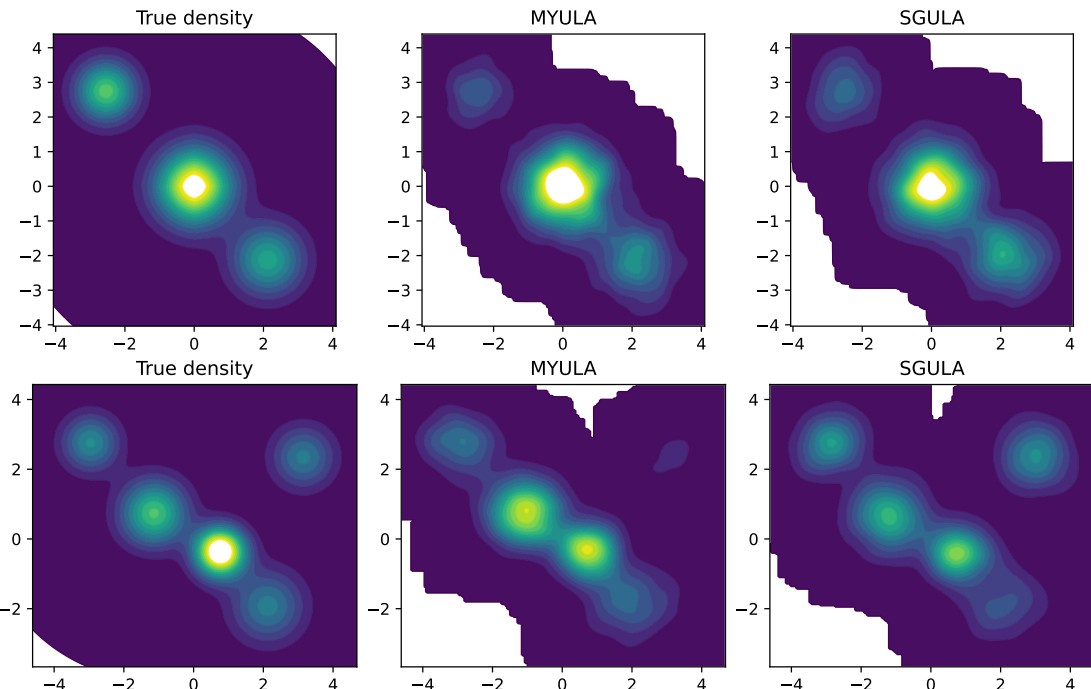

**Figure 1:** Comparison of SGULA and MYULA on a two dimensional mixture of Gaussians target with a Laplace prior. The top row corresponds to case $K = 3$, and the bottom row to $K = 5$.

Note that the subgradient of $u$ grows at most linearly and, in view of Remark 2, it is strongly convex at infinity. Moreover, $K_2 = 16$ and $\mu_1 = 4$.

### 4.3 Multidimensional example satisfying the assumptions

We present an example of a non-convex potential that satisfies our assumptions. Let

$$u(x) = \max\{|x|, |x|^2\} - \frac{1}{2}|x|^2, \quad x \in \mathbb{R}^d.$$

It is easy to see that $u$ is semi-convex (therefore satisfies Assumption A4) as $u + \frac{1}{2}|x|^2$ is convex since it is the maximum of two convex functions. In addition, it is clear to see that each subgradient in this example is bounded inside the ball of radius 1, while outside the function is differentiable with $\nabla u(x) = x$ so it satisfies Assumption A1. The proof of Assumption A2 is more lengthy and is postponed to the Appendix, see Remark 2.

### 4.4 The SCAD Penalty

A notable class of non-convex penalties frequently encountered in sparse recovery problems and high-dimensional statistics is the family of folded concave penalties. Among the most well-known is the *Smoothly Clipped Absolute Deviation (SCAD)* penalty, originally introduced by Fan & Li (2001) as a sparsity-inducing regularizer with unbiasedness properties. We show here that it satisfies our standing assumptions, thereby illustrating a semi-convex objective function that is strongly convex at infinity.

Let $a > 2$ and $\gamma > 0$. A key component of the SCAD function is $q_{a,\gamma} : [0, \infty) \to \mathbb{R}$ which is given by

$$\frac{d}{dx} q_{a,\gamma}(x) = \begin{cases} \gamma, & \text{if } x \leq \gamma, \\ \dfrac{a\gamma - x}{a - 1}, & \text{if } \gamma < x \leq a\gamma, \\ 0, & \text{if } x > a\gamma. \end{cases}$$

Integrating and selecting constants to ensure continuity, we obtain the function $q_{a,\gamma}$,

$$q_{a,\gamma}(x) = \begin{cases} \gamma x, & \text{if } x \leq \gamma, \\ \dfrac{-x^2 + 2a\gamma x - \gamma^2}{2(a-1)}, & \text{if } \gamma < x \leq a\gamma, \\ \dfrac{(a+1)\gamma^2}{2}, & \text{if } x > a\gamma. \end{cases}$$

For any $x \in \mathbb{R}$, one extends the above function by defining $p_{a,\gamma}(x) = q_{a,\gamma}(|x|)$. The resulting function is continuous, symmetric, and non-convex but $1/(2(a-1))$-semi-convex. Its derivative is discontinuous at the origin, reflecting the model's sparsity bias. Further, we define the regularized function $p_{a,\gamma}^r(x) := p_{a,\gamma}(x) + \frac{1}{2(a-1)}x^2$, which is convex. Choosing a subdifferential version that accounts for the discontinuity at zero, one has

$$\partial p_{a,\gamma}^r(0) \in [-\gamma, \gamma] \quad \text{and} \quad \partial p_{a,\gamma}^r(x) = \begin{cases} \dfrac{\gamma x}{|x|} + \dfrac{x}{a-1}, & \text{if } 0 < |x| \leq \gamma, \\ \dfrac{a\gamma x}{(a-1)|x|}, & \text{if } \gamma < |x| \leq a\gamma, \\ \dfrac{x}{a-1}, & \text{if } |x| > a\gamma. \end{cases}$$

A careful case-by-case comparison confirms the monotonicity property $\langle \partial p_{a,\gamma}^r(x) - \partial p_{a,\gamma}^r(y), x - y \rangle \geq 0$ for all $x, y \in \mathbb{R}$. In the multidimensional case, for $x \in \mathbb{R}^d$, we consider the separable extension

$$P_{a,\gamma}(x) := \sum_{i=1}^{d} p_{a,\gamma}(x_i). \tag{12}$$

Then $P_{a,\gamma}$ is also $1/(2(a-1))$-semi-convex, since the regularized form

$$P_{a,\gamma}^r(x) := P_{a,\gamma}(x) + \frac{1}{2(a-1)}|x|^2$$

is convex by separability and convexity of each $p_{a,\gamma}^r$. Indeed, for all $x, y \in \mathbb{R}^d$ and $s \in [0,1]$,

$$\begin{aligned} P_{a,\gamma}^r(sx + (1-s)y) = \sum_{i=1}^{d} p_{a,\gamma}^r(sx_i + (1-s)y_i) &\leq \sum_{i=1}^{d} \left[ s p_{a,\gamma}^r(x_i) + (1-s) p_{a,\gamma}^r(y_i) \right] \\ &= s P_{a,\gamma}^r(x) + (1-s) P_{a,\gamma}^r(y). \end{aligned}$$

Moreover, the subgradient of $P_{a,\gamma}$ is a bounded function. Therefore, by Remark 2, any objective function of the form $u(x) = v(x) + P_{a,\gamma}(x)$, where $v$ is strongly convex (for instance a quadratic), satisfies Assumptions A2–A3. This example highlights how non-convex but semi-convex structures, arising in high-dimensional regularization problems, fall within the scope of our framework. Such penalties are particularly relevant in sparse estimation, compressed sensing, and machine learning applications where both model simplicity and robustness are sought.

### 4.4.1 Robust Regression

To compliment the theoretical analysis and illustrate the applicability of the Subgradient Unadjusted Langevin Algorithm (SGULA) to a practical optimization problem, we consider both a robust regression task with the non-convex SCAD regularization and, for comparison, the standard convex LASSO regularization. By evaluating SCAD and LASSO under an identical optimizer, we demonstrate that non-convex penalties can be handled within the same framework and the theoretical design translates into measurable performance gains.

In this experiment we generate 100 datasets according to the following procedure. Let $x \in \mathbb{R}^d$ with Toeplitz covariance $\Sigma_{ij} = \rho^{|i-j|}$ and $\rho = 0.5$. For a fixed observations $n = 60$ and dimension $d = 8$, we sample $X \in R^{n \times d}$ from the standard Gaussian. The response follows the model $Y = X^T \beta^* + \epsilon$, where

$\beta^* = (3, 1.5, 0, 0, 2, 0, 0, 0)^T$ and the noise is drawn from a heavy tailed mixture, i.e. $\epsilon \sim 0.9\mathcal{N}(0, 1) + 0.1\mathrm{Cauchy}(0, 1)$.

The objective is to minimize the penalized least squares for the SCAD and LASSO regularizations, which yields the potentials $U_S(\beta) = |y - X\beta|^2 + P_{\alpha,\gamma}(\beta)$ and $U_L(\beta) = |y - X\beta|^2 + \gamma|\beta|_1$. Here, $P_{\alpha,\gamma}(\beta)$ denotes the SCAD penalty, with fixed $a = 3.7$, as suggested by Fan & Li (2001). The stepsize is fixed at $\lambda = 10^{-3}$ and the tuning parameter $\gamma > 0$, is chosen independently for both objectives via 5-fold Cross-Validation. Each chain is run for $7.5 \times 10^3$ iterations, while each CV-fold is truncated at $1.25 \times 10^3$ iterations.

Across $R = 100$ Monte Carlo replications, corresponding to the generated dataset, we compute the model error $\mathrm{ME}(\hat{\beta}) = (\hat{\beta} - \beta^*)^T C (\hat{\beta} - \beta^*)$ and the replication wise relative model error $\mathrm{RME} = \mathrm{ME}(\hat{\beta})/\mathrm{ME}(\hat{\beta}_{\mathrm{OLS}})$, reporting the median values, i.e. MRME. We also track the oracle for reference.

Figure 2 displays the MRME boxplots for SCAD, LASSO, and the oracle. Over 100 replications, we observe that SGULA combined with SCAD achieved a median relative model error of 34% , compared to 63% for the LASSO and 29% for the oracle. These results confirm that the non-convex SCAD penalty yields near oracle accuracy under the same subgradient unadjusted Langevin dynamics, providing empirical evidence that SGULA can perform effective excess-risk minimization on semi-convex and non-smooth problems.

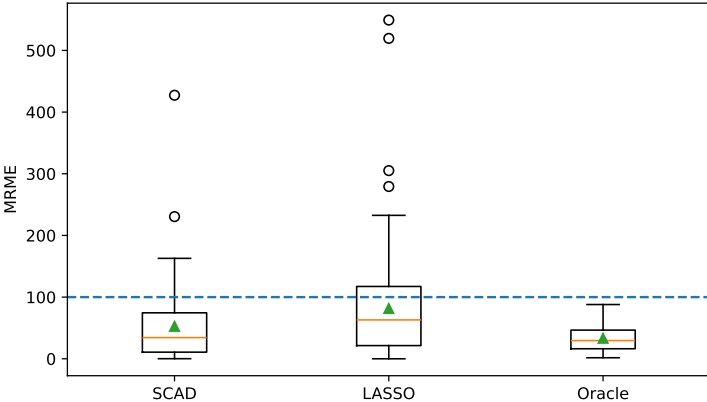

**Figure 2:** Distribution of median relative model errors (MRME %) over 100 Monte Carlo replications for robust regression using SGULA

## 5 Conclusion and discussion

In this work, we have given non-asymptotic guarantees to sample from a target density where the potential is non-convex and not smooth using an algorithm that is simple, computationally efficient, explicit and does not rely on smoothing techniques. Even though, our assumptions are quite relaxed compared to the current literature due to assuming only semi-logconcavity, we establish non-asymptotic guarantees in Wasserstein distances that are comparable to the current state of the art results available in the literature. In addition, we show that our algorithm can also perform well as an optimizer to solve associated (expected) excess-risk optimization problems.

We believe that our current work represents a step forward in bridging the gap in the literature regarding sampling from non-smooth and non-logconcave potentials. Interesting directions for future research include relaxing the assumptions even further and deriving estimates in stronger metrics, such as Rényi divergence, which are useful for differential privacy.

## Impact Statement

This paper presents work aimed at advancing the field of machine learning in the direction of non-convex optimization and associated sampling problems in the presence of discontinuities. While there are many potential societal consequences of our work, we do not believe any require specific emphasis here.

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

## A  Auxiliary Processes

Consider the $\mathbb{R}^d$-valued overdamped Langevin SDE $(Z_t)_{t\in\mathbb{R}_+}$ given by

$$dZ_t = -h(Z_t)dt + \sqrt{2\beta^{-1}}dB_t, \ t \geq 0, \tag{13}$$

with $Z_0 := \theta_0$, where $h \in \partial U$ and $(B_t)_{t\geq 0}$ is a standard $d$-dimensional Brownian motion. To avoid the issue of having set values SDEs, when we use continuous time arguments, we have the convention that at points where $u$ is not differentiable, $h$ is the subgradient with the minimum norm (we can always find it since the set of subgradients is compact and convex). We next introduce the auxiliary processes which are used in our analysis. For each $\lambda > 0$, the time-scaled process $(Z_t^\lambda)_{t\in\mathbb{R}_+}$ is defined by $Z_t^\lambda := Z_{\lambda t}$, $t \in \mathbb{R}_+$. We note that

$$dZ_t^\lambda = -\lambda h(Z_t^\lambda)dt + \sqrt{2\lambda\beta^{-1}}d\tilde{B}_t^\lambda, \ Z_0^\lambda = \theta_0, \tag{14}$$

where the Brownian motion $(\tilde{B}_t^\lambda)_{t\geq 0}$ is defined as $\tilde{B}_t^\lambda := B_{\lambda t}/\sqrt{\lambda}$, $t \geq 0$. The natural filtration of $(\tilde{B}_t^\lambda)_{t\geq 0}$ is denoted by $(\mathcal{F}_t^\lambda)_{t\geq 0}$ with $\mathcal{F}_t^\lambda := \mathcal{F}_{\lambda t}, t \in \mathbb{R}_+$. Then, we define $(\bar{\theta}_t^\lambda)_{t\in\mathbb{R}_+}$, the continuous-time interpolation of SG-ULA (7), as

$$d\bar{\theta}_t^\lambda = -\lambda h(\bar{\theta}_{\lfloor t \rfloor}^\lambda)dt + \sqrt{2\lambda\beta^{-1}}d\tilde{B}_t^\lambda, \ \bar{\theta}_0^\lambda = \theta_0. \tag{15}$$

The law of this process coincides with the law of the algorithm at grid points i.e. $\mathcal{L}(\bar{\theta}_n^\lambda) = \mathcal{L}(\theta_n^\lambda)$ for every $n \in \mathbb{N}$. Furthermore, consider a continuous-time process $(\zeta_t^{s,u,\lambda})_{t\geq s}$, which denotes the solution of the SDE

$$d\zeta_t^{s,u,\lambda} = -\lambda h(\zeta_t^{s,u,\lambda})dt + \sqrt{2\lambda\beta^{-1}}d\tilde{B}_t^\lambda, \ \zeta_s^{s,u,\lambda} = u \in \mathbb{R}^d. \tag{16}$$

**Definition 2.** *Fix $n \in \mathbb{N}$. For any $t \geq nT$, define $\bar{\zeta}_t^{\lambda,n} := \zeta_t^{nT,\bar{\theta}_{nT}^\lambda,\lambda}$, where $T := \lfloor 1/\lambda \rfloor$.*

One notices that the process $(\bar{\zeta}_t^{\lambda,n})_{t\geq nT}$ has the same law as the time-scaled Langevin SDE (14), started at time $nT$ with initial condition $\bar{\theta}_{nT}^\lambda$.

## B  Existence and uniqueness of solution to the SDE and the invariant measure

Consider the infinitesimal generator $\mathcal{L}$ associated with (13) defined for all $\phi \in C^2(\mathbb{R}^d)$ and $x \in \mathbb{R}^d$ by $\mathcal{L}\phi(x) = -\langle h(x), \nabla\phi(x)\rangle + \beta^{-1}\Delta\phi(x)$. Next define the Lyapunov function $V(x) = 1 + |x|^2$ for all $x \in \mathbb{R}^d$. Note that $V$ is twice continuously differentiable, and under Assumption A1, one gets the following growth condition

$$\mathcal{L}V(x) \leq C_*V(x), \ \forall x \in \mathbb{R}^d, \tag{17}$$

where $C_* = \max\{4L, m^2/L\} + m^2/2L + 2d/\beta$. Moreover under both Assumptions A1 and A2, it satisfies the geometric drift condition

$$\mathcal{L}V(x) \leq -\mu V(x) + \mu + 2b + 2d/\beta, \ \forall x \in \mathbb{R}^d. \tag{18}$$

It follows that

$$\lim_{|x|\to\infty} V(x) = +\infty, \ \lim_{|x|\to\infty} \mathcal{L}V(x) = -\infty. \tag{19}$$

**Proposition 1.** *Let Assumptions A1-A4 hold. The SDE (13) has a unique strong solution.*

*Proof.* Uniqueness is guaranteed under the monotonicity condition A4 and due to the diffusion coefficient being constant. Moreover, all conditions of Theorem 2.8 in (Gyöngy & Krylov, 2022) are satisfied under our assumptions; therefore, the SDE (13) admits a unique strong solution. In particular, since the drift coefficient is subject to the growth Assumption A1 and the diffusion coefficient is constant, in view of (17), they trivially satisfy the conditions (i), (ii) and (iv). Condition (iii) is also satisfied trivially as in our case the domain is $D = \mathbb{R}^d$. $\square$

**Proposition 2.** *Let Assumptions A1, A2 and A4 hold, the Langevin SDE* (13) *admits a unique invariant measure.*

*Proof.* The existence of an invariant measure is established under Assumptions A1 and A2. In particular, the Langevin SDE (13) has a constant diffusion coefficient and Assumption A1 ensures that the drift coefficient is locally integrable. Consequently, in view of (19), all the conditions of Theorem 2.2 in (Bogachev et al., 2000) are satisfied, the existence of at least one invariant measure follows. Moreover, with the inclusion of Assumption A4, the contraction results in Appendix C imply the uniqueness of the invariant measure. This is a direct consequence of either Proposition 4 or Proposition 5, by setting the initial condition $Z_0$ in (13) to be such that $\mathcal{L}(Z_0) = \mathcal{L}(\pi_\beta)$. $\qquad\square$

**Proposition 3.** *Let Assumptions A1, A2 and A4 hold. The invariant measure $\pi_\beta$ of the SDE* (13)*, is characterized by the density $Z^{-1} \exp(-\beta u(x))$, with $Z$ being the normalization constant.*

*Proof.* Under Assumption A1 one yields that $u \in \mathbb{H}^1_{\text{loc}}$ and the rest follow from Theorem 3 in Fruehwirth & Habring (2024). $\qquad\square$

**Remark 2.** *Since the dissipativity condition is still preserved when one replaces Assumption A2 with A5, Propositions 1,2, 2 still hold under Assumptions A1, A3, A4, A5.*

## C   Preliminary Estimates

**Lemma 3.** *Let Assumptions A1,A3 and A2 or A5 hold. Then one has*

$$\sup_{t \geq 0} \mathbb{E}|Z_t|^2 \leq C_1 \left(1 + \mathbb{E}|\theta_0|^2\right), \tag{20}$$

*where $C_1 = (4/\mu)(b + d/\beta)$.*

*Proof.* Let $\tau_R = \inf\{t \geq 0 : |Z_t| \geq R\}$. Then by applying Itô's formula to $(t,x) \rightarrow e^{\mu t/2}|x|^2$, one obtains

$$e^{\mu(t \wedge \tau_R)/2}|Z_{t \wedge \tau_R}|^2 = |\theta_0|^2 + \int_0^{t \wedge \tau_R} \frac{\mu}{2} e^{\mu s/2}|Z_s|^2 - 2e^{\mu s/2}\langle Z_s, h(Z_s)\rangle + \frac{2d}{\beta} e^{\mu s/2} ds$$

$$+ \int_0^{t \wedge \tau_R} \sqrt{8\beta^{-1}} e^{\mu s/2} h(Z_s) dB_s.$$

Due to the boundedness of $h$ under Assumption A1, the last term is a martingale, thus vanishing under expectation. Hence by taking the expectation on both sides and using (4), we bound the LHS as follows

$$\mathbb{E}\left[e^{\mu(t \wedge \tau_R)/2}|Z_{t \wedge \tau_R}|^2\right] \leq \mathbb{E}|\theta_0|^2 + \frac{4}{\mu}(b + d/\beta)e^{\mu(t \wedge \tau_R)/2} - \frac{\mu}{2}\int_0^{t \wedge \tau_R} e^{\mu s/2}\mathbb{E}|Z_s|^2 ds$$

$$\leq \mathbb{E}|\theta_0|^2 + \frac{4}{\mu}(b + d/\beta)e^{\mu t/2}.$$

Note furthermore that since $Z_t$ has almost surely continuous trajectories one has $\sup_{s \in [0,t]} |Z_s| < \infty$ (a.s), so by Fatou's Lemma

$$e^{\mu t/2}\mathbb{E}\left[|Z_t|^2\right] = \mathbb{E}\left[\liminf_{R \to \infty} e^{\mu(t \wedge \tau_R)/2}|Z_{t \wedge \tau_R}|^2\right] \leq \liminf_{R \to \infty} \mathbb{E}\left[e^{\mu(t \wedge \tau_R)/2}|Z_{t \wedge \tau_R}|^2\right]$$

$$\leq \mathbb{E}|\theta_0|^2 + \frac{4}{\mu}(b + d/\beta)e^{\mu t/2}.$$

Hence by multiplying both sides by $e^{-\mu t/2}$, we yield the desired result

$$\mathbb{E}\left[|Z_t|^2\right] \leq \mathbb{E}|\theta_0|^2 + \frac{4}{\mu}(b + d/\beta).$$

$\qquad\square$

**Lemma 4.** *Let Assumptions A1-A3 hold and $\lambda_0 \in (0, \mu/(2L^2))$. Then there exists $C_2 > 0$ such that for every $\lambda \in (0, \lambda_0)$ one has*

$$\sup_{t \geq 0} \mathbb{E}|\bar{\theta}_t^\lambda|^2 \leq C_3 \left(1 + \mathbb{E}|\theta_0|^2\right), \tag{21}$$

*where $C_3 = \left(2\mu^2/L^2 + 2\right)C_2 + (2\mu/L^2)\left(\mu m^2/L^2 + 2d/\beta\right)$ and $C_2 = (2b + 2d/\beta + \mu m^2/L^2)/(\mu - 2\lambda L^2)$.*

*Proof.* We begin by considering the SG-ULA iterates $(\theta_n^\lambda)_{n \geq 0}$ (7) corresponding to interpolation scheme (15).

$$|\theta_{n+1}^\lambda|^2 = \left|\theta_n^\lambda - \lambda h(\theta_n^\lambda)\right|^2 + \frac{2\lambda}{\beta}|\xi_{n+1}|^2 + 2\langle \theta_n^\lambda - \lambda h(\theta_n^\lambda), \xi_{n+1}\rangle.$$

Since $\theta_n^\lambda$ is independent of $\xi_{n+1}$, the last term vanishes under expectation. Thus by taking the conditional expectation $\mathbb{E}^{\theta_n^\lambda}[\cdot]$, on both sides and using (2), (4), we obtain

$$\begin{aligned}
\mathbb{E}^{\theta_n^\lambda}\left[|\theta_{n+1}^\lambda|^2\right] &= \mathbb{E}^{\theta_n^\lambda}\left[|\theta_n^\lambda|^2\right] - 2\lambda\mathbb{E}^{\theta_n^\lambda}\left[\langle \theta_n, h(\theta_n^\lambda)\rangle\right] + \lambda^2\mathbb{E}^{\theta_n^\lambda}\left[|h(\theta_n^\lambda)|^2\right] + 2\lambda d/\beta \\
&\leq |\theta_n^\lambda|^2 - \lambda\mu|\theta_n^\lambda|^2 + 2\lambda^2 L^2|\theta_n^\lambda|^2 + 2\lambda b + 2\lambda^2 m^2 + 2\lambda d/\beta \\
&\leq \left(1 - \lambda\mu + 2\lambda^2 L^2\right)|\theta_n^\lambda|^2 + \lambda\left(2b + 2\mu m^2/(2L^2) + 2d/\beta\right).
\end{aligned}$$

Now by taking the expectation on both sides, we can iterate the above bound, due to the restriction $\lambda < \mu/(2L^2)$, to get

$$\begin{aligned}
\mathbb{E}\left[|\theta_{n+1}^\lambda|^2\right] &\leq \left(1 - \left(\lambda\mu - 2\lambda^2 L^2\right)\right)^n \mathbb{E}|\theta_0|^2 \\
&\quad + \frac{1 - \left(1 - \left(\lambda\mu - 2\lambda^2 L^2\right)\right)^n}{\lambda\left(\mu - 2\lambda L^2\right)}\lambda\left(2b + 2d/\beta + \mu m^2/L^2\right) \\
&\leq C_2\left(1 + \mathbb{E}|\theta_0|^2\right). \tag{22}
\end{aligned}$$

For the interpolated scheme, by Hölder's inequality and the linear growth condition (2) one writes

$$\begin{aligned}
|\bar{\theta}_t^\lambda|^2 &= 2|\bar{\theta}_t^\lambda - \bar{\theta}_{\lfloor t \rfloor}^\lambda|^2 + 2|\bar{\theta}_{\lfloor t \rfloor}^\lambda|^2 \leq 4\left|\int_{\lfloor t \rfloor}^t \lambda h(\bar{\theta}_{\lfloor s \rfloor}^\lambda)ds\right|^2 + \frac{8\lambda}{\beta}|d\tilde{B}_t^\lambda - d\tilde{B}_{\lfloor t \rfloor}^\lambda|^2 + 2|\bar{\theta}_{\lfloor t \rfloor}^\lambda|^2 \\
&\leq 4\lambda^2(t - \lfloor t \rfloor)\int_{\lfloor t \rfloor}^t |h(\bar{\theta}_{\lfloor s \rfloor}^\lambda)|^2 ds + \frac{8\lambda}{\beta}|d\tilde{B}_t - d\tilde{B}_{\lfloor t \rfloor}|^2 + 2|\bar{\theta}_{\lfloor t \rfloor}^\lambda|^2 \\
&\leq 4\lambda^2 \int_{\lfloor t \rfloor}^t (2m^2 + 2L^2|\bar{\theta}_{\lfloor s \rfloor}^\lambda|^2)ds + \frac{8\lambda}{\beta}|d\tilde{B}_t - d\tilde{B}_{\lfloor t \rfloor}|^2 + 2|\bar{\theta}_{\lfloor t \rfloor}^\lambda|^2.
\end{aligned}$$

Notice that for any $s \in [\lfloor t \rfloor, t]$, we have $\lfloor s \rfloor = \lfloor t \rfloor$, thus by taking the expectation we obtain

$$\mathbb{E}|\bar{\theta}_t^\lambda|^2 \leq 8\lambda^2 m^2 + \frac{8\lambda d}{\beta} + \left(8\lambda^2 L^2 + 2\right)\mathbb{E}|\bar{\theta}_{\lfloor t \rfloor}^\lambda|^2 \leq \frac{2\mu}{L^2}\left(\frac{\mu m^2}{L^2} + \frac{2d}{\beta}\right) + \left(\frac{2\mu^2}{L^2} + 2\right)\mathbb{E}|\bar{\theta}_{\lfloor t \rfloor}^\lambda|^2.$$

Moreover, by construction the interpolation scheme (15) agrees with the SG-ULA iterates (7) on grid points. That is $\bar{\theta}_{\lfloor t \rfloor}^\lambda = \theta_n^\lambda$ for $t \in [n, n+1)$, thus by using the bound established in (22), we yield

$$\mathbb{E}|\bar{\theta}_t^\lambda|^2 \leq C_3\left(1 + \mathbb{E}|\theta_0|^2\right).$$

$\square$

**Lemma 5.** *Let Assumptions A1-A3 hold and $\lambda_0 \in (0, \mu/(2L^2))$. Then there exists $C_4 > 0$ such that for every $\lambda \in (0, \lambda_0)$ one has*

$$\mathbb{E}|\bar{\theta}_{\lfloor t \rfloor}^\lambda - \bar{\theta}_t^\lambda|^2 \leq C_4\lambda(1 + \mathbb{E}|\theta_0|^2), \tag{23}$$

*where $C_4 = 2\mu C_3 + 2\mu m^2/L^2 + 4d/\beta$.*

*Proof.* One considers the difference between $\bar{\theta}^\lambda_{\lfloor t \rfloor}, \bar{\theta}^\lambda_t$ to get the one-step error

$$|\bar{\theta}^\lambda_{\lfloor t \rfloor} - \bar{\theta}^\lambda_t|^2 \le 2 \left| \int_{\lfloor t \rfloor}^t \lambda h(\bar{\theta}^\lambda_{\lfloor s \rfloor}) ds \right|^2 + \frac{4\lambda}{\beta} \left| \tilde{B}^\lambda_{\lfloor t \rfloor} - \tilde{B}^\lambda_t \right|^2.$$

Taking the expectation and applying Hölder's inequality, the linear growth condition (2) and Lemma 4, yield

$$\mathbb{E}|\bar{\theta}^\lambda_{\lfloor t \rfloor} - \bar{\theta}^\lambda_t|^2 \le 2\lambda^2 \int_{\lfloor t \rfloor}^t \left( 2m^2 + 2L^2 C_3 \left(1 + \mathbb{E}|\theta_0|^2\right)\right) ds + 4\lambda d/\beta$$
$$\le \lambda^2 \left(2m^2 + 2L^2 C_3 + 2L^2 C_3 \mathbb{E}|\theta_0|^2\right) + 4\lambda d/\beta.$$

$\square$

**Lemma 6.** *Let Assumptions A1-A3 hold and $\lambda_0 \in (0, \mu/(2L^2))$. Then there exists $C_5 > 0$, such that for every $\lambda \in (0, \lambda_0)$ and $n \in \mathbb{N}$, one has*

$$\sup_{nT \le t \le (n+1)T} \mathbb{E}|\bar{\zeta}^{\lambda,n}_t|^2 \le C_5(1 + \mathbb{E}|\theta_0|^2), \tag{24}$$

*where $C_5 = C_3 + 2(d/\beta + b)$.*

*Proof.* Standard arguments show that one has boundedness enough that the stochastic integral vanishes,, we obtain the existence of a constant c, which depends on time, such that $\sup_{t \ge nT} \mathbb{E}|\bar{\zeta}^{\lambda,n}_t|^2 \le c < \infty$. Furthermore, by applying Itô's formula and taking expectations one has

$$\mathbb{E}|\bar{\zeta}^{\lambda,n}_t|^2 = \mathbb{E}|\bar{\theta}^\lambda_{nT}|^2 - \int_{nT}^t \lambda \mathbb{E}\langle h(\bar{\zeta}^{\lambda,n}_s), \bar{\zeta}^{\lambda,n}_s\rangle ds + 2\lambda d\beta^{-1}(t - nT).$$

Then, differentiating both sides and using (4)

$$\frac{d}{dt}\mathbb{E}|\bar{\zeta}^{\lambda,n}_t|^2 \le -\lambda\mu\mathbb{E}|\bar{\zeta}^{\lambda,n}_s|^2 + 2\lambda(d/\beta + b)$$
$$\frac{d}{dt}e^{\lambda\mu(t-nT)}\mathbb{E}|\bar{\zeta}^{\lambda,n}_t|^2 \le 2\lambda(d/\beta + b)e^{\lambda\mu(t-nT)}$$
$$\mathbb{E}|\bar{\zeta}^{\lambda,n}_t|^2 \le e^{-\lambda\mu(t-nT)}\mathbb{E}|\bar{\theta}^\lambda_{nT}|^2 + 2\lambda(t - nT)(d/\beta + b).$$

Due to $nT \le t \le (n+1)T$ and in view Lemma 4 one gets

$$\mathbb{E}|\bar{\zeta}^{\lambda,n}_t|^2 \le C_3(1 + \mathbb{E}|\theta_0|^2) + 2(d/\beta + b).$$

$\square$

**Lemma 7.** *Let Assumptions A1-A4 hold and $\lambda_0 \in (0, \mu/(2L^2))$. Then there exists $C_6 > 0$, such that for every $\lambda \in (0, \lambda_0)$, $n \in \mathbb{N}$ and $t \in [nT, (n+1)T]$, one obtains*

$$W_2(\mathcal{L}(\bar{\theta}^\lambda_t), \mathcal{L}(\bar{\zeta}^{\lambda,n}_t)) \le C_6 \lambda^{1/4},$$

*where $C_6 = \sqrt{2}e^{2K} \left(C_4(1 + \mathbb{E}|\theta_0|^2)\right)^{1/4} \sqrt{\sqrt{C_4(1 + \mathbb{E}|\theta_0|^2)} + 2L \left(1 + \sqrt{C_5(1 + \mathbb{E}|\theta_0|^2)} + \sqrt{C_2(1 + \mathbb{E}|\theta_0|^2)}\right)}$. The same result holds if one replaces Assumption A2 with A5.*

*Proof.* In order to bound the $W_2$ distance it suffices to bound $\mathbb{E}|\bar{\theta}_t^\lambda - \bar{\zeta}_t^{\lambda,n}|^2$ where these processes are solutions to SDEs with same initial condition and same Brownian motion. Applying Itô's formula one obtains

$$
\begin{aligned}
\mathbb{E}|\bar{\theta}_t^\lambda - \bar{\zeta}_t^{\lambda,n}|^2 &= -2\lambda \int_{nT}^t \mathbb{E}\langle \bar{\theta}_s^\lambda - \bar{\zeta}_s^{\lambda,n}, h(\bar{\theta}_{\lfloor s \rfloor}^\lambda) - h(\bar{\zeta}_s^{\lambda,n})\rangle ds \\
&= -2\lambda \int_{nT}^t \mathbb{E}\langle \bar{\theta}_{\lfloor s \rfloor}^\lambda - \bar{\zeta}_s^{\lambda,n}, h(\bar{\theta}_{\lfloor s \rfloor}^\lambda) - h(\bar{\zeta}_s^{\lambda,n})\rangle ds - 2\lambda \int_{nT}^t \mathbb{E}\langle \bar{\theta}_s^\lambda - \bar{\theta}_{\lfloor s \rfloor}^\lambda, h(\bar{\theta}_{\lfloor s \rfloor}^\lambda) - h(\bar{\zeta}_s^{\lambda,n})\rangle ds \\
&\leq 2\lambda K \int_{nT}^t \mathbb{E}|\bar{\theta}_{\lfloor s \rfloor}^\lambda - \bar{\zeta}_s^{\lambda,n}|^2 ds + 2\lambda \mathbb{E}\int_{nT}^t |\bar{\theta}_s^\lambda - \bar{\theta}_{\lfloor s \rfloor}^\lambda||h(\bar{\theta}_{\lfloor s \rfloor}^\lambda) - h(\bar{\zeta}_s^{\lambda,n})| ds \\
&\leq 4\lambda K \int_{nT}^t \mathbb{E}|\bar{\theta}_s^\lambda - \bar{\zeta}_s^{\lambda,n}|^2 ds \\
&\quad + 4\lambda K \int_{nT}^t \mathbb{E}|\bar{\theta}_{\lfloor s \rfloor}^\lambda - \bar{\theta}_s^\lambda|^2 ds + 2\lambda \mathbb{E}\int_{nT}^t |\bar{\theta}_s^\lambda - \bar{\theta}_{\lfloor s \rfloor}^\lambda||h(\bar{\theta}_{\lfloor s \rfloor}^\lambda) - h(\bar{\zeta}_s^{\lambda,n})| ds,
\end{aligned}
$$

where the first inequality was obtained using the one-sided Lipschitz Assumption A4. The second term can be controlled by Lemma 6 while for the third term we apply Hölder's inequality with $\epsilon = 1$,

$$
\begin{aligned}
\mathbb{E}\int_{nT}^t |\bar{\theta}_s^\lambda - \bar{\theta}_{\lfloor s \rfloor}^\lambda||h(\bar{\theta}_{\lfloor s \rfloor}^\lambda) - h(\bar{\zeta}_s^{\lambda,n})| ds &\leq \int_{nT}^t \left( \mathbb{E}|\bar{\theta}_s^\lambda - \bar{\theta}_{\lfloor s \rfloor}^\lambda|^{1+\epsilon}\right)^{1/(1+\epsilon)} \left( \mathbb{E}|h(\bar{\theta}_{\lfloor s \rfloor}^\lambda) - h(\bar{\zeta}_s^{\lambda,n})|^{(1+\epsilon)/\epsilon}\right)^{\epsilon/(1+\epsilon)} ds \\
&\leq \int_{nT}^t \sqrt{\mathbb{E}|\bar{\theta}_s^\lambda - \bar{\theta}_{\lfloor s \rfloor}^\lambda|^2}\sqrt{2L^2 \mathbb{E}(1 + |\bar{\zeta}_s^{\lambda,n}|^2 + |\bar{\theta}_{\lfloor s \rfloor}^\lambda|^2)} ds \\
&\leq 2LT\sqrt{C_4(1 + \mathbb{E}|\theta_0|^2)} \\
&\quad \times \left( 1 + \sqrt{C_5(1 + \mathbb{E}|\theta_0|^2)} + \sqrt{C_2(1 + \mathbb{E}|\theta_0|^2)}\right)\lambda^{1/2},
\end{aligned}
\tag{25}
$$

where the inequality follows from the Cauchy-Schwarz inequality, the second uses the linear growth property of the gradient, (Assumption A1) and the final bound is obtained using estimates in Lemma 5, along with the moment bounds of the algorithm and the auxiliary process provided in Lemmata 4 and 6 respectively. Putting all together, leads to

$$
\mathbb{E}|\bar{\theta}_t^\lambda - \bar{\zeta}_t^{\lambda,n}|^2 \leq 4\lambda K \int_{nT}^t \mathbb{E}|\bar{\theta}_s^\lambda - \bar{\zeta}_s^{\lambda,n}|^2 ds + 2C\lambda^{1/2},
$$

where $C = 2C_4(1 + \mathbb{E}|\theta_0|^2) + \sqrt{C_4(1 + \mathbb{E}|\theta_0|^2)}2L\left(1 + \sqrt{C_5(1 + \mathbb{E}|\theta_0|^2)} + \sqrt{C_2(1 + \mathbb{E}|\theta_0|^2)}\right)$. Since the right hand side is finite (as there is a control of the moments of the algorithm and the auxiliary process at finite time), one can apply Grönwall's inequality which yields

$$
\mathbb{E}|\bar{\theta}_t^\lambda - \bar{\zeta}_t^{\lambda,n}|^2 \leq 2e^{4K}C\lambda^{1/2}.
$$

Since $W_2(\mathcal{L}(\bar{\theta}_t^\lambda), \mathcal{L}(\bar{\zeta}_t^{\lambda,n})) \leq \sqrt{\mathbb{E}|\bar{\theta}_t^\lambda - \bar{\zeta}_t^{\lambda,n}|^2}$ the result follows immediately. □

**Remark 3.** *According to the choice made in (25), setting $\epsilon = 1$, leads to a discretization error of order $\lambda^{1/4}$ in $W_2$ distance. More generally, taking arbitrary $\epsilon > 0$ in the Hölder step produces a bound of the form $C_\epsilon \lambda^{1/(2+2\epsilon)}$ which approaches the classical $\lambda^{1/2}$ rate when $\epsilon$ is small. The constant $C_\epsilon$ deteriorates as $\epsilon$ decreases, in particular through a stronger dependence on the dimension. For this reason, we work with $\epsilon = 1$ which results in a milder dependence on the dimension.*

## D  Contraction Estimates

**Proposition 4.** *Let Assumptions A1-A4 hold. Consider $Z_t', t \geq 0$, be the solution of (13) with initial condition $Z_0' = \theta_0'$, which is independent of $\mathcal{F}_\infty$ and satisfies Assumption A3. Then*

$$
W_1\left(\mathcal{L}(Z_t), \mathcal{L}(Z_t')\right) \leq C_{W_1}e^{-C_{r_1}t}W_1(\mathcal{L}(\theta_0), \mathcal{L}(\theta_0')),
\tag{26}
$$

where $C_{W_1} = 2e^{\beta K R^2/8}$ and $C_{r_1} = 2\beta^{-1} C_0'$ with

$$
C_0' = \begin{cases}
\dfrac{2}{3e} \min(1/R^2, \mu\beta/8) & \text{if } \beta K R^2 \leq 8, \\[2mm]
\left(8\sqrt{2\pi} R^{-1} (\beta K)^{-1/2}((\beta K)^{-1} + (\beta\mu)^{-1}) \exp(\beta K R^2/8) + 32(\beta\mu R)^{-2}\right)^{-1} & \text{if } \beta K R^2 \geq 8.
\end{cases}
$$

*Proof.* It follows directly by invoking Theorem 1, Corollary 2 and Lemma 1 in (Eberle, 2016). □

**Proposition 5.** *Let Assumptions A1-A4 hold. Consider $Z_t', t \geq 0$, be the solution of (13) with initial condition $Z_0' = \theta_0'$, which is independent of $\mathcal{F}_\infty$ and satisfies Assumption A3. Then, for any $\epsilon \in (0, \sqrt{\beta/8\mu})$*

$$
W_2\left(\mathcal{L}(Z_t), \mathcal{L}(Z_t')\right) \leq C_{W_2} e^{-C_{r_2} t} \max\left\{ W_2(\mathcal{L}(\theta_0), \mathcal{L}(\theta_0')), \sqrt{W_1(\mathcal{L}(\theta_0), \mathcal{L}(\theta_0'))} \right\}, \tag{27}
$$

*where $C_{W_2} = 2\max\{1, R^{-1/2}\} C_0''(\epsilon) e^{\left(\sqrt{\beta/32}(\mu+K)+\epsilon/2\right)\beta R^2/2} \sqrt{(2/\beta)\max\{4/\epsilon + 2, 8/(e\epsilon^2)\}/(\sqrt{\beta/2}R + 1)}$, $C_{r_2} = 2\min\{1, 1/\epsilon\} e^{-(1/4)\sqrt{(\beta/2)^3}(\mu+K)R^2}/C_0''(\epsilon)$, and $C_0''(\epsilon)$ depends exclusively on $\beta\mu$ and can be found in Table 2.*

*Proof.* It follows directly by invoking Theorem 1.3 in (Luo & Wang, 2016). □

**Proposition 6.** *Let Assumptions A1, A3, A4, A5 hold. Consider $Z_t', t \geq 0$, be the solution of (13) with initial condition $Z_0' = \theta_0'$, which is independent of $\mathcal{F}_\infty$ and satisfies Assumption A3. Then*

$$
W_2\left(\mathcal{L}(Z_t), \mathcal{L}(Z_t')\right) \leq C_{W_2}^* e^{-C_{r_3} t} W_2(\mathcal{L}(Z_0), \mathcal{L}(Z_0')), \tag{28}
$$

*where $C_{W_2}^* = \sqrt{1 + (2d)^{-1}\beta(2K + \mu)(2 + 2K/\mu)^{2/d}}$ and $C_{r_3} = \mu/4$.*

*Proof.* It follows directly by invoking Theorem 1 in (Monmarché, 2023). □

**Lemma 8.** *Let Assumptions A1-A4 hold and $\lambda_0 \in (0, \mu/(2L^2))$. Then there exists $C_7 > 0$, such that for every $\lambda \in (0, \lambda_0)$, $n \in \mathbb{N}$ and $t \in [nT, (n+1)T]$, one obtains*

$$
W_1(\mathcal{L}(\bar{\zeta}_t^{\lambda,n}), \mathcal{L}(Z_t^\lambda)) \leq C_7 \lambda^{1/4},
$$

*where $C_7 = C_6 C_{W_1}/\left(1 - e^{-C_{r_1}/2}\right)$.*

*Proof.* Recall that $\mathcal{L}(Z_t^\lambda) = \mathcal{L}(\zeta_t^{\lambda,0})$ so using the triangle inequality for the Wasserstein distance one deduces that

$$
\begin{aligned}
W_1(\mathcal{L}(\bar{\zeta}_t^{\lambda,n}), \mathcal{L}(Z_t^\lambda)) &\leq \sum_{k=1}^n W_1\left(\mathcal{L}\left(\bar{\zeta}_t^{\lambda,k}\right), \mathcal{L}\left(\bar{\zeta}_t^{\lambda,k-1}\right)\right) \\
&= \sum_{k=1}^n W_1\left(\mathcal{L}\left(\zeta_t^{kT, \bar{\theta}_{kT}^\lambda, \lambda}\right), \mathcal{L}\left(\zeta_t^{(k-1)T, \bar{\theta}_{(k-1)T}^\lambda, \lambda}\right)\right) \\
&= \sum_{k=1}^n W_1\left(\mathcal{L}\left(\zeta_t^{kT, \bar{\theta}_{kT}^\lambda, \lambda}\right), \mathcal{L}\left(\zeta_t^{kT, \bar{\zeta}_{kT}^{\lambda,k-1}, \lambda}\right)\right) \\
&\leq C_{W_1} \sum_{k=1}^n \exp(-C_{r_1}(n-k)\lambda T) W_1\left(\mathcal{L}\left(\bar{\theta}_{kT}^\lambda\right), \mathcal{L}\left(\bar{\zeta}_{kT}^{\lambda,k-1}\right)\right) \\
&\leq C_{W_1} \sum_{k=1}^n \exp(-C_{r_1}(n-k)\lambda T) W_2\left(\mathcal{L}\left(\bar{\theta}_{kT}^\lambda\right), \mathcal{L}\left(\bar{\zeta}_{kT}^{\lambda,k-1}\right)\right),
\end{aligned}
$$

where in the first two equalities we used the definition 2 of auxiliary process and the in the last inequalities we applied the contraction property in Proposition 4 and the fact that $W_1 \leq W_2$. This further implies, due to $\lambda T = \lambda \lfloor 1/\lambda \rfloor \in (1/2, 1]$ and the discretization error estimates from Lemma 7

$$
W_1(\mathcal{L}(\bar{\zeta}_t^{\lambda,n}), \mathcal{L}(Z_t^\lambda)) \leq C_{W_1} \frac{1}{1 - e^{-C_{r_1}/2}} C_6 \lambda^{1/4}.
$$

$\square$

**Lemma 9.** *Let Assumptions A1-A4 hold hold and $\lambda_0 \in (0, \mu/(2L^2))$. Then there exists $C_8 > 0$, such that for every $\lambda \in (0, \lambda_0)$, $n \in \mathbb{N}$ and $t \in [nT, (n+1)T]$, one obtains*

$$W_2(\mathcal{L}(\bar{\zeta}_t^{\lambda,n}), \mathcal{L}(Z_t^\lambda)) \leq C_8 \lambda^{1/8},$$

*where $C_8 = \max\{C_6, \sqrt{C_6}\} C_{W_2} / \left(1 - e^{-C_{r_2}/2}\right)$.*

*Proof.* Recall that $\mathcal{L}(Z_t^\lambda) = \mathcal{L}(\zeta_t^{\lambda,0})$ so using the triangle inequality for the Wasserstein distance one deduces that

$$
\begin{aligned}
W_2(\mathcal{L}(\bar{\zeta}_t^{\lambda,n}), \mathcal{L}(Z_t^\lambda)) &\leq \sum_{k=1}^n W_2\left(\mathcal{L}\left(\bar{\zeta}_t^{\lambda,k}\right), \mathcal{L}\left(\bar{\zeta}_t^{\lambda,k-1}\right)\right) \\
&= \sum_{k=1}^n W_2\left(\mathcal{L}\left(\zeta_t^{kT, \bar{\theta}_{kT}^\lambda, \lambda}\right), \mathcal{L}\left(\zeta_t^{(k-1)T, \bar{\theta}_{(k-1)T}^\lambda, \lambda}\right)\right) \\
&= \sum_{k=1}^n W_2\left(\mathcal{L}\left(\zeta_t^{kT, \bar{\theta}_{kT}^\lambda, \lambda}\right), \mathcal{L}\left(\zeta_t^{kT, \bar{\zeta}_{kT}^{\lambda,k-1}, \lambda}\right)\right) \\
&\leq C_{W_2} \sum_{k=1}^n \exp(-C_{r_2}(n-k)\lambda T) \\
&\quad \times \max\left\{W_2\left(\mathcal{L}\left(\bar{\theta}_{kT}^\lambda\right), \mathcal{L}\left(\bar{\zeta}_{kT}^{\lambda,k-1}\right)\right), \sqrt{W_1\left(\mathcal{L}\left(\bar{\theta}_{kT}^\lambda\right), \mathcal{L}\left(\bar{\zeta}_{kT}^{\lambda,k-1}\right)\right)}\right\},
\end{aligned}
$$

where the first two equalities are deduced by the definition of the auxiliary process and the last inequality by the contraction property in Proposition 5. This further implies, due to $\lambda T = \lambda \lfloor 1/\lambda \rfloor \in (1/2, 1]$ and the discretization error estimates from Lemma 7

$$
\begin{aligned}
W_2(\mathcal{L}(\bar{\zeta}_t^{\lambda,n}), \mathcal{L}(Z_t^\lambda)) &\leq C_{W_2} \sum_{k=1}^n \exp(\frac{C_{r_2}}{2}(n-k)) \max\{C_6, \sqrt{C_6}\} \lambda^{1/8} \\
&\leq C_{W_2} \frac{1}{1 - e^{-C_{r_2}/2}} \max\{C_6, \sqrt{C_6}\} \lambda^{1/8}.
\end{aligned}
$$

$\square$

**Lemma 10.** *Let Assumptions A1, A3, A4, A5 and $\lambda_0 \in (0, \mu/(2L^2))$. Then, there exists $C_9 > 0$, such that for every $\lambda < \lambda_0$, $n \in \mathbb{N}$ and $t \in (nT, (n+1)T]$, one obtains*

$$W_2(\mathcal{L}(\bar{\zeta}_t^{\lambda,n}), \mathcal{L}(Z_t^\lambda)) \leq C_9 \lambda^{1/4},$$

*where $C_9 = C_6 C_{W_2}^* / (1 - e^{-C_{r_3}/2})$.*

*Proof.* The proof is similar to the previous ones, the difference being that we use an improved contraction result of Proposition 6

$$
\begin{aligned}
W_2(\mathcal{L}(\bar{\zeta}_t^{\lambda,n}), \mathcal{L}(Z_t^\lambda)) &\le \sum_{k=1}^n W_2\left(\mathcal{L}\left(\bar{\zeta}_t^{\lambda,k}\right), \mathcal{L}\left(\bar{\zeta}_t^{\lambda,k-1}\right)\right) \\
&= \sum_{k=1}^n W_2\left(\mathcal{L}\left(\zeta_t^{kT,\bar{\theta}_{kT}^\lambda,\lambda}\right), \mathcal{L}\left(\zeta_t^{(k-1)T,\bar{\theta}_{(k-1)T}^\lambda,\lambda}\right)\right) \\
&= \sum_{k=1}^n W_2\left(\mathcal{L}\left(\zeta_t^{kT,\bar{\theta}_{kT}^\lambda,\lambda}\right), \mathcal{L}\left(\zeta_t^{kT,\bar{\zeta}_{kT}^{\lambda,k-1},\lambda}\right)\right) \\
&\le C_{W_2}^* \sum_{k=1}^n \exp(-C_{r_3}(n-k)\lambda T) W_2\left(\mathcal{L}(\bar{\theta}_{kT}^\lambda), \mathcal{L}\left(\bar{\zeta}_{kT}^{\lambda,k-1}\right)\right) \\
&\le C_{W_2}^* \sum_{k=1}^n \exp(-\frac{C_{r_3}}{2}(n-k)) C_6 \lambda^{1/4} \\
&= C_{W_2}^* \frac{1}{1-e^{-C_{r_3}/2}} C_6 \lambda^{1/4}.
\end{aligned}
$$

$\square$

## E    Estimates for the excess risk optimization problem

**Lemma 11.** *Let Assumptions A1-A4 hold and $\lambda_0 \in (0, \mu/(2L^2))$. Then, for every $\lambda \in (0, \lambda_0)$ and $n \in \mathbb{N}$, the following bound for $\mathcal{T}_1 = \mathbb{E}[u(\theta_n^\lambda)] - \mathbb{E}[u(\theta_\infty)]$ holds*

$$
\mathcal{T}_1 \le C_{\mathcal{T}_1} W_2(\mathcal{L}(\theta_n^\lambda), \pi_\beta), \tag{29}
$$

*where $C_{\mathcal{T}_1} = m + (L/2)\sqrt{\mathbb{E}|\theta_0|^2} + (L/2)\sqrt{C_\sigma}$ and $C_\sigma = (\mu + 2b + 2d/\beta)/\mu$.*

*Proof.* We notice that the function $g(t) = u(tx + (1-t)y)$ is locally Lipschitz continuous (since $u$ is semi-convex) so it has a bounded variation in $[0,1]$. Then, one can enforce the fundamental theorem of calculus since $g'(t) = \langle h(tx + (1-t)y), x - y \rangle$ a.e. Thus, one writes

$$
\begin{aligned}
u(x) - u(y) &= \int_0^1 \langle x - y, h((1-t)y + tx) \rangle dt \le \int_0^1 |x-y||h((1-t)y + tx)| dt \\
&\le \int_0^1 |x-y|(m + L|(1-t)y + tx|) dt \le (m + (L/2)|x| + (L/2)|y|)|x-y|, \tag{30}
\end{aligned}
$$

where we have used Cauchy-Schwarz and the growth Assumption A1. Now let $(X, Y)$ be the coupling of $\mu$, $\nu$ that achieves $W_2(\mu, \nu)$, that is $W_2^2(\mu, \nu) = \mathbb{E}|X - Y|^2$ for $\mathcal{L}(X) = \mu$ and $\mathcal{L}(Y) = \mu$. Taking expectations in (30) and using Minkowski's inequality, yields

$$
\begin{aligned}
\int_{\mathbb{R}^d} g d\mu - \int_{\mathbb{R}^d} g d\nu &= \mathbb{E}[g(X) - g(Y)] \le \sqrt{\mathbb{E}[(m + (L/2)|x| + (L/2)|y|)^2]} \sqrt{\mathbb{E}|X-Y|^2} \\
&\le \left(m + (L/2)\sqrt{\mathbb{E}|X|^2} + (L/2)\sqrt{\mathbb{E}|Y|^2}\right) W_2(\mu, \nu). \tag{31}
\end{aligned}
$$

One concludes by applying inequality (31) for $X = u(\theta_n^\lambda)$ and $Y = u(\theta_\infty)$

$$
\mathbb{E}[u(\theta_n^\lambda)] - \mathbb{E}[u(\theta_\infty)] \le \left(m + (L/2)\sqrt{\mathbb{E}|\theta_0|^2} + (L/2)\sqrt{C_\sigma}\right) W_2(\mathcal{L}(\theta_n^\lambda), \pi_\beta), \tag{32}
$$

where $C_\sigma$ is the second-moment of $\pi_\beta$. Since $\pi_\beta$ is the invariant measure of SDE (13), there holds $\int_{\mathbb{R}^d} \mathcal{L}V(x)\pi_\beta(dx) = 0$. Due to (18), one estimates the constant by

$$
C_\sigma \le \int_{\mathbb{R}^d} V(x)\pi_\beta(dx) \le -\mu \int_{\mathbb{R}^d} \mathcal{L}V(x)\pi_\beta(dx) + (\mu + 2b + 2d/\beta)/\mu \le (\mu + 2b + 2d/\beta)/\mu. \tag{33}
$$

$\square$

**Lemma 12.** *Let Assumptions A1-A4 hold. For any $\beta \geq \max\{4/\mu, M^{-1}\}$, the following bound for $\mathcal{T}_2 = \mathbb{E}[u(\theta_\infty)] - u(\theta^*)$ holds*

$$\mathcal{T}_2 \leq \frac{d}{2\beta} \log \left( \frac{2e(b + d/\beta)\beta^2 M^2}{\mu d} \right) + \frac{2}{\beta} - \frac{1}{\beta} \log(S_d/d) + \frac{d}{\beta} \log(\beta M). \tag{34}$$

*where the associated constants are given explicitly in the proof.*

*Proof.* We follow a similar approach as in Section 3.5 of Raginsky et al. (2017), making necessary adjustments due to the lack of a smoothness condition for the gradient $\nabla u(x) := h(x)$. According to Raginsky et al. (2017), one obtains the following bound

$$\mathbb{E}[u(\theta_\infty)] \leq \frac{d}{2\beta} \log \left( \frac{4\pi e(b + d/\beta)}{\mu d} \right) - \frac{1}{\beta} \log Z, \tag{35}$$

where $Z := \int_{\mathbb{R}^d} e^{-\beta u(x)} dx$ is the normalization constant. One writes

$$\log Z = \log \int_{\mathbb{R}^d} e^{-\beta u(x)} dx = -\beta u(\theta^*) + \log \int_{\mathbb{R}^d} e^{\beta(u(\theta^*) - u(x))} dx. \tag{36}$$

Now we provide an upper bound for the second term of (36). For the remainder of this analysis, one chooses the version of the subgradient $h(x)$ such that $h(\theta^*) = 0$. Therefore, property (4) immediately implies $|\theta^*| \leq \sqrt{b/(2\mu)} = R_2$. Consequently, one calculates that

$$-(u(\theta^*) - u(x)) \leq |u(\theta^*) - u(x)| \leq \int_0^1 |\langle h(x + t(\theta^* - x)), \theta^* - x \rangle|\, dt \leq \int_0^1 |h(x + t(\theta^* - x))||\theta^* - x| dt$$

$$\leq \int_0^1 (m + L|x| + tL|\theta^* - x|) |\theta^* - x| dt$$

$$\leq \int_0^1 (m + L|\theta^* - x| + L|\theta^*| + tL|\theta^* - x|) |\theta^* - x| dt$$

$$\leq (3L/2)|\theta^* - x|^2 + (m + LR_2)|\theta^* - x|.$$

Hence we obtain

$$I = \int_{\mathbb{R}^d} e^{\beta(u(\theta^*) - u(x))} dx \geq \int_{\mathbb{R}^d} e^{-\beta(3L/2)|\theta^* - x|^2 - \beta(m + LR_2)|\theta^* - x|} dx = \int_{\mathbb{R}^d} e^{-\beta M(|\theta^* - x| + |\theta^* - x|^2)} dx \tag{37}$$

where $M = 3L/2 + m + LR_2$. Changing to radial coordinates one obtains

$$I = S_d \int_0^\infty e^{-\beta M(r + r^2)} r^{d-1} dr$$

where $S_d = 2\pi^{d/2} \Gamma^{-1}(d/2)$. Assuming that $\beta M \geq 1$,

$$I \geq S_d \int_0^{(\beta M)^{-1}} e^{-\beta M(r + r^2)} r^{d-1} dr \geq S_d \int_0^{(\beta M)^{-1}} e^{-2} r^{d-1} dr = S_d d^{-1} e^{-2} (\beta M)^{-d}.$$

Combining the aforementioned bounds with equation (36) leads to

$$\frac{1}{\beta} \log Z \geq -u(\theta^*) - \frac{1}{\beta} \log(2) + \frac{1}{\beta} \log(S_d/d) - \frac{d}{\beta} \log(\beta M).$$

In view of (30), one concludes with

$$\mathbb{E}[u(\theta_\infty)] - u(\theta^*) \leq \frac{d}{2\beta} \log \left( \frac{2e(b + d/\beta)\beta^2 M^2}{\mu d} \right) + \frac{2}{\beta} - \frac{1}{\beta} \log(S_d/d) + \frac{d}{\beta} \log(\beta M).$$

$\square$

# F   Proofs of Section 3

**Proof of Theorem 1**

*Proof.* Let $N \in \mathbb{N}$ and set $n = \lfloor N/T \rfloor$, then $N \in [nT, (n+1)T]$. Fix $\lambda \in (0, \lambda_0)$, and $t \in [nT, (n+1)T]$. Then, for $p \in \{1, 2\}$, by the triangle inequality,

$$W_p\left(\mathcal{L}(\theta_N^\lambda), \pi_\beta\right) \leq W_p\left(\mathcal{L}(\bar{\theta}_N^\lambda), \mathcal{L}(\bar{\zeta}_N^{\lambda,n})\right) + W_p\left(\mathcal{L}(\bar{\zeta}_N^{\lambda,n}), \mathcal{L}(Z_N^\lambda)\right) + W_p\left(\mathcal{L}(Z_N^\lambda), \pi_\beta\right).$$

By Lemmata 7-9 and Propositions 4-5, these three terms satisfy

$$W_p\left(\mathcal{L}(\bar{\theta}_N^\lambda), \mathcal{L}(\bar{\zeta}_N^{\lambda,n})\right) \leq W_2\left(\mathcal{L}(\bar{\theta}_N^\lambda), \mathcal{L}(\bar{\zeta}_N^{\lambda,n})\right) \leq C_6 \lambda^{1/4},$$

$$W_1\left(\mathcal{L}(\bar{\zeta}_N^{\lambda,n}), \mathcal{L}(Z_N^\lambda)\right) \leq C_7 \lambda^{1/4} \text{ and } W_2\left(\mathcal{L}(\bar{\zeta}_N^{\lambda,n}), \mathcal{L}(Z_N^\lambda)\right) \leq C_8 \lambda^{1/8},$$

$$W_p\left(\mathcal{L}(Z_N^\lambda), \pi_\beta\right) \leq C_{W_p} e^{-C_{r_p} \lambda N} \Delta_0^{(p)},$$

where

$$\Delta_0^{(1)} = W_1\left(\mathcal{L}(\theta_0), \pi_\beta\right), \quad \Delta_0^{(2)} = \max\left\{W_2\left(\mathcal{L}(\theta_0), \pi_\beta\right), \sqrt{W_1\left(\mathcal{L}(\theta_0), \pi_\beta\right)}\right\}.$$

Combining the three bounds yields, for $p \in \{1, 2\}$,

$$W_1\left(\mathcal{L}(\theta_N^\lambda), \pi_\beta\right) \leq C_{W_1} e^{-C_{r_1} \lambda N} \Delta_0^{(1)} + (C_6 + C_7)\lambda^{1/4},$$

$$W_2\left(\mathcal{L}(\theta_N^\lambda), \pi_\beta\right) \leq C_{W_2} e^{-C_{r_2} \lambda N} \Delta_0^{(2)} + (C_6 + C_8)\lambda^{1/8}.$$

Setting $C_{T_1} = C_6 + C_7$ and $C_{T_2} = C_6 + C_9$, gives the final statement. $\qquad \square$

**Proof of Theorem 2**

*Proof.* Let $N \in \mathbb{N}$ and set $n = \lfloor N/T \rfloor$, then $N \in [nT, (n+1)T]$. Therefore, taking into account the results of Lemmata 7, 10 and Proposition 6, it follows that for every $\lambda \in (0, \lambda_0)$, $n \in \mathbb{N}$, and $t \in [nT, (n+1)T]$, one has

$$W_2\left(\mathcal{L}(\theta_N^\lambda), \pi_\beta\right) \leq W_2\left(\mathcal{L}(\bar{\theta}_N^\lambda), \mathcal{L}(\bar{\zeta}_N^{\lambda,n})\right) + W_2\left(\mathcal{L}(\bar{\zeta}_N^{\lambda,n}), \mathcal{L}(Z_N^\lambda)\right) + W_2\left(\mathcal{L}(Z_N^\lambda), \pi_\beta\right)$$

$$\leq C_{W_2}^* e^{-C_{r_3} \lambda N} W_2(\mathcal{L}(\theta_0), \pi_\beta) + (C_6 + C_9)\lambda^{1/4}.$$

$$\square$$

**Proof of Theorem 3**

*Proof.* Fix $n \in \mathbb{N}, \lambda \in (0, \lambda_0)$, and $\beta \geq \max\{4/\mu, M^{-1}\}$. Consider an independent draw $\theta_\infty \sim \pi_\beta$, then one decomposes the excess risk error

$$\mathbb{E}\left[u(\theta_n^\lambda)\right] - u(\theta^*) = \left(\mathbb{E}\left[u(\theta_n^\lambda)\right] - \mathbb{E}\left[u(\theta_\infty)\right]\right) + \left(\mathbb{E}\left[u(\theta_\infty)\right] - u(\theta^*)\right) := \mathcal{T}_1 + \mathcal{T}_2.$$

Under Assumptions A1-A4, using the inequalities from Lemmata 11-12, yields the final bound. $\qquad \square$

# G   Auxiliary Remarks

**Proof of Lemma 2**

*Proof.* Let $|x| \geq R$, then through A2 one obtains

$$\langle x, h(x) \rangle = \langle x - 0, h(x) - h(0) \rangle + \langle x, h(0) \rangle \geq \mu |x|^2 - |x||h(0)| \geq \frac{\mu}{2}|x|^2 - \frac{|h(0)|}{2\mu}. \tag{38}$$

Now let $|x| < R$, due to the linear growth in A1 one writes

$$\langle x, h(x) \rangle \geq -|x||h(x)| \geq -m|x| - L|x|^2 \geq -mR - LR^2 + \frac{\mu}{2}R^2 - \frac{\mu}{2}R^2$$

$$\geq \frac{\mu}{2}|x|^2 - \left(mR + (L + \mu/2)R^2\right). \tag{39}$$

Combining (38) and (39) yield (4), where $b = \max\left(|h(0)|/(2\mu), mR + (L + \mu/2)R^2\right)$. $\qquad \square$

The following Remark is a useful tool to verify A2 when a function is known to be strongly convex outside a compact set but not necessarily inside of it.

**Remark 4.** *Let $R > 0$ and suppose $u(x) \in C(\mathbb{R}^d)$ and is given by $u(x) = \begin{cases} u_1(x), |x| \leq R \\ u_2(x), |x| > R \end{cases}$ , where $u_1, u_2 : \mathbb{R}^d \to \mathbb{R}$ admit the gradients $h_1 = \nabla u_1$, $h_2 = \nabla u_2$ such that*

$$h_{1,2,R} = \max \left\{ \sup_{|x| \leq R} |h_1(x)|, \sup_{|x| \leq R} |h_2(x)| \right\} < \infty.$$

*Moreover, $u_2$ is $\mu$-strongly convex. Then $u$ is $\mu/2$-strongly convex at infinity, outside the ball $\mathcal{B}\left(0, (2\sqrt{2}/\mu)h_{1,2,R}\right)$.*

*Proof.* Let $x \in \mathcal{B}(0, R)$ and $y \notin \mathcal{B}(0, R)$, one writes

$$\begin{aligned}
\langle x - y, h(x) - h(y) \rangle &= \langle x - y, h_1(x) - h_2(y) = \langle x - y, h_2(x) - h_2(y) \rangle + \langle x - y, h_1(x) - h_2(x) \rangle \\
&\geq \mu |x - y|^2 - |x - y||h_1(x) - h_2(x)| \\
&\geq \mu |x - y|^2 - (\mu/4)|x - y|^2 - (1/\mu)|h_1(x) - h_2(x)|^2 \\
&\geq (3\mu/4)|x - y|^2 - (2/\mu)h_{1,2,R}^2 = (3\mu/4)|x - y|^2 - (\mu/4)\bar{R}^2.
\end{aligned}$$

Hence for any $x, y$ such that $|x - y| > \bar{R} = (2\sqrt{2}/\mu)h_{1,2,R}$, one obtains

$$\langle x - y, h(x) - h(y) \rangle \geq (\mu/2)|x - y|^2.$$

$\qquad \square$

# H   Complimentary numerical experiments

This section provides additional simulation results expanding upon Subsection 4.1.1, illustrating the behavior of SGULA under varying stepsizes and inverse temperature parameters in the same Gaussian mixture with Laplacian prior setting.

First we explore the behavior of SGULA across a range of stepsizes: $\lambda = \{10^{-1}, 10^{-2}, 10^{-3}, 10^{-4}, 10^{-5}\}$. In Figure 3 we observe that a moderate stepsize ($\lambda = 10^{-3}$) yields the most faithful approximation to the true target density, successfully recovering the modal structure and allocating mass in accordance with the true distribution. For larger stepsizes, while the major modes are still detected, some modes appear overrepresented while others are suppressed, indicating that the sampler suffers from discretization bias. This is consistent with the well known behavior of Langevin based samplers, large stepsizes cause the discrete time dynamics to deviate from the continuous time Langevin diffusion, leading to biased stationary distributions. Conversely, when using smaller stepsizes, the shape of the empirical distribution deteriorates despite correct mode localization. The samples appear fragmented or noisy, with reduced mass between modes. This occurs because smaller stepsizes slow down the exploration of the space, leading to poor mixing

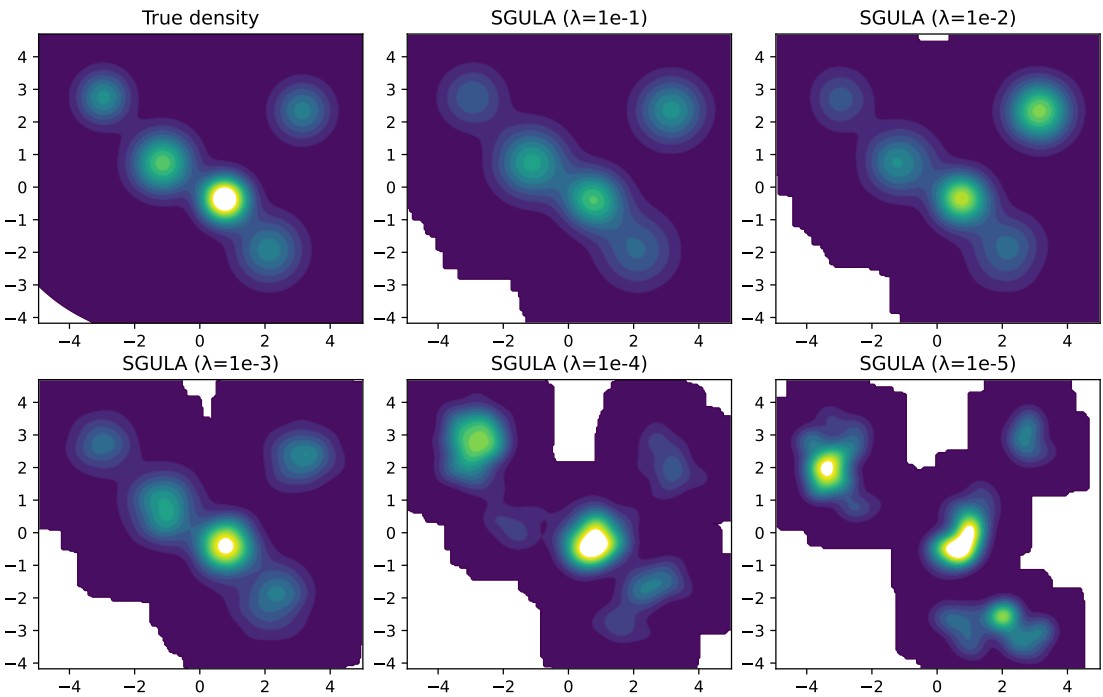

**Figure 3:** SGULA applied to a two-dimensional Gaussian mixture with Laplace prior, for varying stepsizes.

and high autocorrelation between samples. As a result, the Markov chains may not adequately transition across modes within the finite computational budget, even if the theoretical bias is small.

Subsequently we run the same experiment for a fixed stepsize $\lambda = 10^{-3}$ and varying inverse temperature parameters $\beta = \{100, 100, 5, 2, 1\}$. In Figure 4 we observe that as $\beta$ increases, the sampler increasingly concentrates around local maxima of the original density. This results in sharp, isolated regions of mass but many modes become underrepresented or completely missing. This behavior is desirable in optimization contexts (e.g., MAP estimation), where identifying a single mode is sufficient, but it undermines full posterior exploration in Bayesian settings. This phenomenon is well understood in the context of Langevin-type algorithms as increasing $\beta$ steepens the potential. As a result, the sampler rapidly descends into local minima and becomes metastable, i.e., it takes exponentially long to escape a mode, especially in multimodal or semi-convex targets.

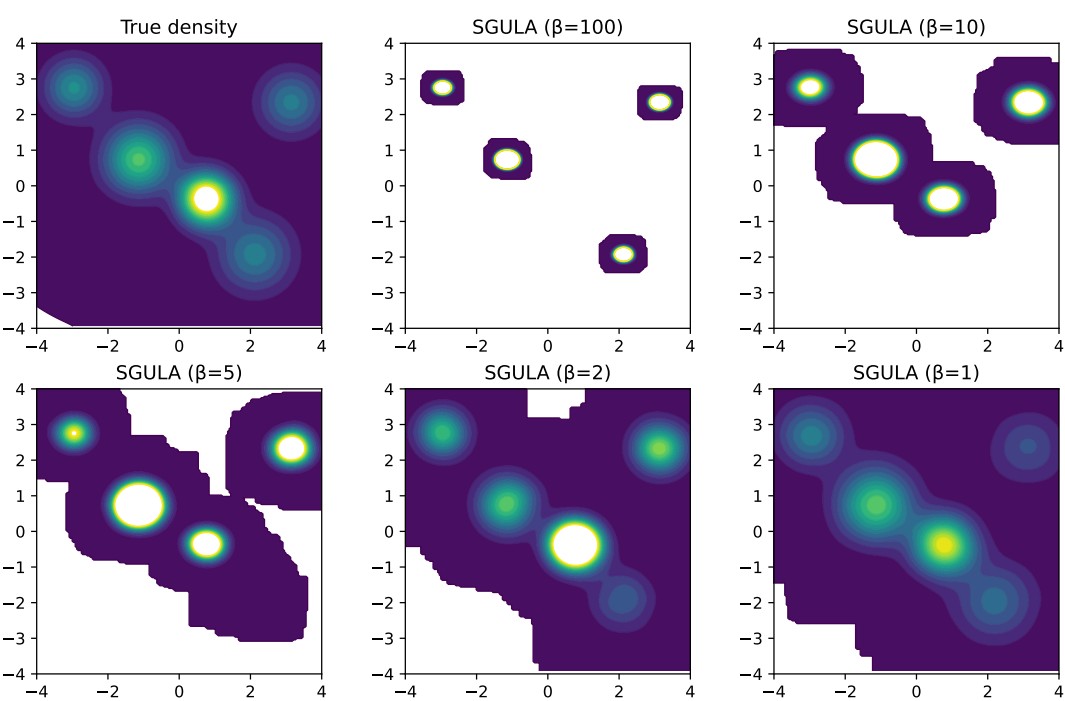

**Figure 4:** SGULA applied to a two-dimensional Gaussian mixture with Laplace prior, for varying inverse temperature parameters.

**Table 2:** Analytic expressions of constants.

| No. | Constants | dim |
|---|---|---|
| 1 | $b = \max\left(|h(0)|/(2\mu), mR + (L + (\mu/2))R^2\right)$ | $\mathcal{O}(1)$ |
| 2 | $C_1 = (4/\mu)(b + d/\beta)$ | $\mathcal{O}(d)$ |
| 3 | $C_2 = (2b + 2d/\beta + \mu m^2/L^2)/(\mu - 2\lambda L^2)$ | $\mathcal{O}(d)$ |
| 4 | $C_3 = \left(2\mu^2/L^2 + 2\right)C_2 + \left(2\mu/L^2\right)\left(\mu m^2/L^2 + 2d/\beta\right)$ | $\mathcal{O}(d)$ |
| 5 | $C_4 = 2\mu C_3 + 2\mu m^2/L^2 + 4d/\beta$ | $\mathcal{O}(d)$ |
| 6 | $C_5 = C_3 + 2(d/\beta + b)$ | $\mathcal{O}(d)$ |
| 7 | $C_6 = e^{2K}\sqrt{C_4(1 + \mathbb{E}|\theta_0|^2)}\left(\sqrt{C_4(1 + \mathbb{E}|\theta_0|^2)} + 2L\left(1 + \sqrt{C_5(1 + \mathbb{E}|\theta_0|^2)} + \sqrt{C_2(1 + \mathbb{E}|\theta_0|^2)}\right)\right)$ | $\mathcal{O}(d)$ |
| 8 | $C_{W_1} = 2e^{\beta K R^2/8}$ | $\mathcal{O}(1)$ |
| 9 | $C_{r_1} = 2\beta^{-1}C_0'$ | $\mathcal{O}(1)$ |
| 10 | $C_0' = \begin{cases} \dfrac{2}{3e}\min(1/R^2, \mu\beta/8) & \text{if } \beta K R^2 \le 8, \\ \left(8\sqrt{2\pi}R^{-1}(\beta K)^{-1/2}((\beta K)^{-1} + (\beta\mu)^{-1})\exp(\beta K R^2/8) + 32(\beta\mu R)^{-2}\right)^{-1} & \text{if } \beta K R^2 \ge 8. \end{cases}$ | $\mathcal{O}(1)$ |
| 11 | $C_{W_2} = 2\max\{1, R^{-1/2}\}C_0''(\epsilon)e^{\left(\sqrt{\beta/32}(\mu+K)+\epsilon/2\right)\beta R^2/2}\sqrt{(2/\beta)\max\{4/\epsilon + 2, 8/(e\epsilon^2)\}}/(\sqrt{\beta/2}R + 1)$ | $\mathcal{O}(1)$ |
| 12 | $C_{r_2} = 2\min\{1, 1/\epsilon\}e^{-(1/4)\sqrt{(\beta/2)^3(\mu+K)R^2}}/C_0''(\epsilon)$ | $\mathcal{O}(1)$ |
| 13 | $C_0''(\epsilon) = \max\left\{\dfrac{2e^2}{\epsilon}\left(1 + \dfrac{2}{\sqrt{\epsilon}}\right)\sqrt{\dfrac{2}{\sqrt{\beta/8}\mu - \epsilon}}, \dfrac{2 + \sqrt{\epsilon}}{\epsilon(1 - e^{-2})}\left[\dfrac{2\sqrt{2}e^2}{\sqrt{\epsilon(\sqrt{\beta/8}\mu - \epsilon)}} + \dfrac{1}{\sqrt{\beta/8}\mu - \epsilon}\right]\right\}$ | $\mathcal{O}(1)$ |
| 14 | $C_{W_2}^* = \sqrt{1 + (2d)^{-1}\beta(2K + \mu)(2 + 2K/\mu)^{2/d}}$ | $\mathcal{O}(d^{-1})$ |
| 15 | $C_{r_3} = \mu/4$ | $\mathcal{O}(1)$ |
| 16 | $C_7 = C_6 C_{W_1}/\left(1 - e^{-C_{r_1}/2}\right)$ | $\mathcal{O}(d)$ |
| 17 | $C_8 = \max\{C_6, \sqrt{C_6}\}C_{W_2}/\left(1 - e^{-C_{r_2}/2}\right)$ | $\mathcal{O}(d)$ |
| 18 | $C_9 = C_6 C_{W_2}^*/\left(1 - e^{-C_{r_3}/2}\right)$ | $\mathcal{O}(d)$ |
| 19 | $C_{T_1} = C_6(1 + C_{W_1}/(1 - e^{-C_{r_1}/2}))$ | $\mathcal{O}(d)$ |
| 20 | $C_{T_2} = C_6 + \max\{C_6, \sqrt{C_6}\}C_{W_2}/\left(1 - e^{-C_{r_2}/2}\right)$ | $\mathcal{O}(d)$ |
| 21 | $C_{T_3} = C_6(1 + C_{W_2}^*/(1 - e^{-C_{r_3}/2}))$ | $\mathcal{O}(d)$ |
| 22 | $C_{\mathcal{T}_1} = m + (L/2)\sqrt{\mathbb{E}|\theta_0|^2} + (L/2)\sqrt{(\mu + 2b + 2d/\beta)/\mu}$ | $\mathcal{O}(d^{1/2})$ |
| 23 | $M = m + 3L/2 + L\sqrt{b/(2\mu)}$ | $\mathcal{O}(1)$ |

