# OpenReview forum: "The Performance Of The Unadjusted Langevin Algorithm Without Smoothness Assumptions"
_TMLR — Accepted by TMLR_

### Review · Reviewer_L6Rv · 2025-08-29

**Summary Of Contributions:**

This paper considers sampling from target distributions with non-smooth and
non-logconcave potential functions. It discusses a few relaxed assumptions
instead of the commonly used smoothness assumptions such as Lipschitz
continuity and differentiability. It then gives examples of potential functions
that satisfy these assumptions. Based on these relaxed assumptions, the paper
derives non-asymptotic bounds on the convergence of the SG-ULA algorithm,
which are claimed to be the first theoretical results without smoothness and
log-concavity assumptions.

The paper appears to be written for experts already familiar with this specific
field and its terminology. I find it difficult to follow.

**Audience:**

Yes

**Audience Explanation:**

Sampling from non-smooth and non-log-concave densities is both challenging and
important. Researchers working on MCMC, computational statistics, and related
areas are likely to be interested in the study.

**Claims And Evidence:**

No

**Claims Explanation:**

My main complaint concerns the writing and organization. The introduction is
clear and presents the motivation and contributions well, although some notation
should be explained more carefully for readers who are not experts in this
field.

Section 2 titled "Theoretical Framework" is mostly a list of assumptions. I
found it hard to see why Section 3 is important as presented. The few examples
showing when the assumptions hold do not appear to serve a clear purpose in the
main text. These examples would be better placed in the appendix so the main
text can focus on the key ideas and results. Relatedly, the paper frequently
refers to the appendix for important notations and for the algorithm itself.
Section 4 discusses "the proposed algorithm" even though the algorithm is not
presented in the main paper but only in the appendix. Important definitions, the
algorithm, and notation that are essential to understanding the results should
appear in the main text.

The setups above equations (11) and (12) appear to be identical. If they are the
same, it is unclear why they are split across Theorems 1 and 2. These should be
unified or the distinction should be explained. Also, remove the ", for more
details see Table 2" from Theorem 2 (or integrate the necessary details into the
statement or its immediate discussion).

The literature review is placed at the end of the paper, which makes it
difficult for readers to assess the paper's contribution relative to existing
work. Consider moving the related work earlier (near the introduction). Section
5.3 is not about literature review.

**Requested Changes:**

Please address the issues noted above.

Here are some additional, more specific comments and suggestions:
1. Explain $Z_t$ and $B_t$ in the displayed equation on page 1.
2. Explain the meaning of the inequality sign in the displayed equation on page 2.
3. The notation $u$ is used for potential functions; consider using a different symbol in Section 1.1.
4. Clarify the meaning of $|·|$ in Definition 1.
5. Assumption numbering is inconsistent (e.g., "Assumption A2" vs "Assumption
   4") .
6. Remark 1 is referenced before it appears. The current presentation seem to
   indicate that it is very important. If so, it is better to state it as a
   lemma or proposition.
7. Define $N$ in equation (6).
8. The proof of Theorem 4 merely refers to two lemmas, which is not helpful for
   readers. If most proofs are deferred, state once in the paper (for example,
   "All proofs are given in the appendix") to save space and avoid repetition.
9. Display equation (15) in the main text rather than only in the appendix.

---

> ### Author Response · Authors · 2025-10-10
>
> Thank you for the feedback, the paper's structure has been revised to improve readability and coherence, the specific modifications are detailed later in the response.
>
> 1. Reviewer: My main complaint concerns the writing and organization. The introduction is clear and presents the motivation and contributions well, although some notation should be explained more carefully for readers who are not experts in this field.
>
> Response: The notation in the Introduction has been clarified and provided with explicit context to remove potential ambiguities.
>
> 2. Reviewer: Section 2 titled "Theoretical Framework" is mostly a list of assumptions.
>
> Response: The title of Section 2 has been changed to "The Non-Convex Setting". This section includes a first subsection, introducing the key notions of subdifferential and weak convexity, which are central to the analysis, and a second subsection outlining the assumptions that characterize the class of problems addressed by SGULA. As the central goal of the paper is to understand why SGULA remains effective beyond the standard assumptions of strong convexity and smoothness, rather than to propose a new algorithm, we consider it essential to present these definitions and assumptions prior to the main theoretical results.
>
> 3. Reviewer:  I found it hard to see why Section 3 is important as presented. The few examples showing when the assumptions hold do not appear to serve a clear purpose in the main text. These examples would be better placed in the appendix so the main text can focus on the key ideas and results.
>
> Response: We also recognized that the previous Section 3 disrupted the logical flow of the paper. It has therefore been relocated to the end of the main body, now forming Section 4, where it is expanded with numerical experiments. Since all reviewers emphasized the importance of including empirical validation, part of these experiments were retained in the main text rather than moved to the appendix.
>
> 4. Reviewer: Relatedly, the paper frequently refers to the appendix for important notations and for the algorithm itself. Section 4 discusses "the proposed algorithm" even though the algorithm is not presented in the main paper but only in the appendix. Important definitions, the algorithm, and notation that are essential to understanding the results should appear in the main text
>
> Response: We removed references to the appendix for material that is central to understanding the method. The algorithmic description is now uniquely placed at the start of Section 3, immediately before the main theorems, where it is invoked.
>
> 5. Reviewer: The setups above equations (11) and (12) appear to be identical. If they are the same, it is unclear why they are split across Theorems 1 and 2. These should be unified or the distinction should be explained. Also, remove the ", for more details see Table 2" from Theorem 2 (or integrate the necessary details into the statement or its immediate discussion).
>
> Response: Theorems 1 and 2 have been merged into a single result, now presented as Theorem 1, to eliminate redundancy. The reference to Table 2 has been removed from the theorem statement and the relevant information has been integrated into the immediate discussion that follows the theorem.
>
> 6. Reviewer: The literature review is placed at the end of the paper, which makes it difficult for readers to assess the paper's contribution relative to existing work. Consider moving the related work earlier (near the introduction). Section 5.3 is not about literature review.
>
> Response: The literature review has been moved to the Introduction as Subsection 2.1. We consider this placement more useful to the reader, as it allows the paper's contributions to be understood in direct relation to existing work from the outset. The previous inconsistency with old Section 5.3 has been resolved.
>
> Requested Changes.
>
> 1. The missing details for the displayed equation on page 1 have been added.
>
> 2. The statement on page 2 has been reformulated without the previously used inequality sign to remove any ambiguity.
>
> 3. We now use x,y instead.
>
> 4. Following also other reviewers observations, Definition 1 has been restated to eliminate ambiguity.
>
> 5. Fixed.
>
> 6. Remark 1 has been renamed to Lemma and relocated to Subsection 2.2 to appear in the proper logical order and align with the presentation of the theoretical framework.
>
> 7. Fixed.
>
> 8. We have retained the deferred proofs, as this structure allows interested readers to navigate the appendix efficiently and access the detailed arguments as needed. The proof of Theorem 4 has been rewritten to include additional details and has been moved to the appendix.
>
> 9. The equation previously labeled as (15) has been removed from the appendix, as noted in the broader discussion of the paper's reorganization. Its content is sufficiently covered within the main text.

---

### Review · Reviewer_U4Ya · 2025-09-14

**Summary Of Contributions:**

In this submission, the authors proved the non-asymptotic guarantees to sample from a target density when the potential is non-convex and not smooth. The authors proposed a novel algorithm that is simple, computationally efficient and didn't require smoothing techniques. The assumptions in this submission are relaxed compared to the previous literature assuming only semi-logconcavity, The authors established non-asymptotic guarantees in Wasserstein distances that are comparable to the current state of the art results available in the literature. The authors also showed that the algorithm can also perform well as an optimizer to solve expected excess-risk optimization problems.

**Audience:**

Yes

**Audience Explanation:**

The authors expand the theory in Langevin-based sampling from non-smooth potentials beyond log-concavity,and semi-logconcavity and to establish connections with non-convex optimization. The authors provide rigorous results for the treatment of SDEs with discontinuous drifts beyond log-concavity.These are interesting theoretical contributions.

**Broader Impact Concerns:**

No concerns on the ethical implications of the work that would require adding a Broader Impact Statement.

**Claims And Evidence:**

Yes

**Claims Explanation:**

The manuscript addresses an important gap in the Langevin-based sampling literature by removing smoothness and logconcavity assumptions. The extension to semi-convex and even discontinuous log-gradients is both timely and significant. However, while the theoretical results are solid, the contribution would be greatly strengthened by empirical validation beyond toy examples (MoG with L1 prior, SCAD). At present, the claims rest almost entirely on theory.

The assumptions (A1–A5) are clearly laid out, but some are stronger than needed for many practical applications. For example, strong convexity at infinity (A2) could be difficult to verify in practice. It would be helpful if the authors explored weaker sufficient conditions or demonstrated empirically that their algorithm performs well even when assumptions are only approximately satisfied.

The literature review is comprehensive, and Table 1 positions the proposed algorithm relative to others. However, the comparison is almost entirely theoretical. Since competing approaches (e.g., MYULA, taming-based ULA) are computationally available, the lack of numerical comparison weakens the practical impact.

The manuscript makes an interesting claim that SG-ULA can solve excess-risk optimization problems. This is theoretically supported, but no optimization experiments are presented. Since the optimization perspective is central to motivating the work, it should be empirically validated.

Overall, this is a strong and rigorous theoretical contribution with real potential impact. However, without numerical experimental validation, the manuscript falls short of demonstrating practical relevance. Reciewer recommend major revisions with the addition of experiments as outlined above, particularly numerical comparisons with MYULA and optimization benchmarks.

If those are provided, the paper would represent a substantial step forward in understanding and applying Langevin algorithms in non-smooth, non-logconcave settings.

**Requested Changes:**

1. The introduction could be streamlined: it currently mixes background, motivation, and contributions in a dense way. A clearer statement of novelty versus prior work would help.

2. Some proofs are postponed to appendices, which is fine, but the main text could include more intuitive explanation of why contraction results hold under semi-convexity.

3. The definition of semi-convexity (Definition 1) should be restated in simpler terms to aid accessibility.

4. Equation numbering is inconsistent: e.g., Algorithm (10) vs. (15). Standardize notation for SG-ULA across the manuscript.

5. Most importantly, the authors need to provide empirical results from numerical experiments to validate the theoretical analysis.

---

> ### Author Response · Authors · 2025-10-10
>
> Thank you for the feedback. We acknowledge the importance of complementing the theoretical analysis with empirical validation. To this end, we have added two numerical experiments: sampling from a mixture of Gaussians with a Laplacian prior (Subsection 4.1.1), and robust regression with SCAD regularization (Subsection 4.4.1). These experiments extend the evaluation beyond toy examples and demonstrate SGULA's behavior across different problem settings.
>
> 1. Reviewer: The assumptions (A1–A5) are clearly laid out, but some are stronger than needed for many practical applications. For example, strong convexity at infinity (A2) could be difficult to verify in practice. It would be helpful if the authors explored weaker sufficient conditions or demonstrated empirically that their algorithm performs well even when assumptions are only approximately satisfied.
>
> Response: We recognize that strong convexity at infinity (A2) can be non-trivial to verify in practice. As discussed in Remark 3 (appendix), this condition can often be checked through more tractable criteria.
>
> 2. Reviewer: The literature review is comprehensive, and Table 1 positions the proposed algorithm relative to others. However, the comparison is almost entirely theoretical. Since competing approaches (e.g., MYULA, taming-based ULA) are computationally available, the lack of numerical comparison weakens the practical impact.
>
> Response: A  numerical comparison with MYULA has been added to the sampling task (Subsection 4.1.1). These results provide a practical evaluation of SGULA relative to established proximal-based algorithms. Regarding taming-based ULA methods, we note that they address challenges of a parallel nature, focusing on stabilizing the dynamics under superlinear drifts, while SGULA targets non-smooth and weakly convex potentials. The two frameworks tackle different sources of instability, they are conceptually extending ULA beyond the classical setting in different ways.
>
> 3. Reviewer: The manuscript makes an interesting claim that SG-ULA can solve excess-risk optimization problems. This is theoretically supported, but no optimization experiments are presented. Since the optimization perspective is central to motivating the work, it should be empirically validated.
>
> Response: To substantiate the optimization perspective of the work, we have added Subsection 4.4.1, which presents a robust regression experiment with SCAD regularization. This example demonstrates the practical  use of SGULA as an optimization algorithm, thereby providing empirical validation of the theoretical claims regarding excess-risk minimization.
>
> Requested changes:
>
> 1. We have moved the literature review and comparison to the introduction, in response to this and related feedback from other reviewers. We believe this revised structure allows readers to more easily identify the key contributions and how they contrast with prior work.
>
> 2. Additional geometric intuition has been added to the discussion following A4, explaining why semi-convexity, in combination with strong convexity at infinity, is sufficient to guarantee the contraction property.
>
> 3. Definition 1 has been restated to remove ambiguities.
>
> 4. This issue, also raided by other reviewers, has been resolved. The equation numbering is now consistent throughout the manuscript, Eq(15), previously in the appendix, has been removed. The SGULA algorithm is now stated once, immediately before the mains results.
>
> 5. Empirical results have been added in Subsections 4.1.1 and 4.4.1 to validate the theoretical analysis. These experiments are discussed in more detail, in the main body of the response, to this and the rest of the reviewers.

---

### Review · Reviewer_hiQn · 2025-09-26

**Summary Of Contributions:**

This work considers analyzing the performance of the unadjusted langevin algorithm for sampling from a distribution whose potential is nonconvex and nonsmooth. In particular, the authors focus on the class of semi-convex, or weakly-convex, potentials. While prior works in the nonsmooth case have focused on smoothing or proximal techniques, this work focuses on the more natural approach of directly using subgradients. The authors derive convergence guarantees in Wasserstein distance for the law of the iterates along with expected convergence guarantees of the iterates to approximate global minimizers of the potential. Such guarantees apply to several interesting examples, including mixtures of Gaussians with a Laplace prior and the SCAD penalty.

**Audience:**

Yes

**Audience Explanation:**

Yes, for example, researchers in sampling and Bayesian inference would find the results interesting.

**Claims And Evidence:**

Yes

**Claims Explanation:**

Yes, the claims made are supported by accurate, convincing and clear evidence. The work tackles an interesting question in using a more simple, direct approach to sample from distributions with potentially nonsmooth potentials while also allowing for nonconvexity without the use of smoothing techniques. The derives convergence guarantees are rigorous and, to the reviewer's knowledge, prior guarantees for such nonsmooth potentials have not been shown previously.

**Requested Changes:**

While not critical to securing my recommendation, I think having simple numerical experiments illustrating convergence of the algorithm with respect to the various parameters (such as the step size $\lambda$ and the temperature $\beta$) would certainly strengthen the paper. It would also help to see, for example, how this simple algorithm fairs to smoothing and proximal techniques along with experimentally how the convergence depends on the above parameters. This would not need to be at the scale of, say, imaging inverse problems with nonsmooth priors, but could be for more modest synthetic data distributions.

---

> ### Author Response · Authors · 2025-10-10
>
> We appreciate the reviewer's feedback and suggestion to complement the theoretical analysis with empirical validation. In response, we have added several numerical experiments in Section 4 that illustrate the convergence of practical performance of SGULA and highlight its relation to existing methods.
>
> In Subsection 4.1.1, we compare SGULA with MYULA on a mixture of Gaussians with Laplacian prior, a setting that aligns with our weakly convex and non-smooth framework. Moreover, in Subsection 4.4.1, we employ SGULA as an optimization method in a robust regression problem regularized with the SCAD penalty, thereby demonstrating its performance in a non-convex and non-smooth optimization task.
>
> Finally, we have added a new appendix (Appendix H) with additional experiments illustrating the behavior of SGULA under varying stepsizes and inverse temperature parameters, using the same setup as in Section  4.1.1. These results confirm known tradeoffs in Langevin-type methods and complement our main findings. If the reviewer believes these experiments would better serve the reader in the main text, we are happy to move them accordingly.

---

### Author Response · Authors · 2025-10-10

We thank the reviewers for their feedback and comments. A revised version of the paper has been uploaded, where the requested changes are highlighted in red.

---

### Decision · Action_Editor_PY8o · 2025-10-28

**Recommendation:** Accept as is

**Audience:**

Yes

**Audience Explanation:**

The algorithm being studied in this work, ULA, can be considered as one of the most fundamental MCMC techniques.

**Claims And Evidence:**

Yes

**Claims Explanation:**

The paper is providing interesting rigorous theoretical results on Unadjusted Langevin Algorithm. The results are supported by rigorous proofs.